

# Aliphatic Carbonyl Compounds (C$_8$-C$_{26}$) in Wintertime Atmospheric Aerosol in London, UK

Ruihe Lyu[1,2], Mohammed Salim Alam[1], Christopher Stark[1]

Ruixin Xu[1], Zongbo Shi[2], Yinchang Feng[2] and Roy M. Harrison[†∗1]

[1] Division of Environmental Health and Risk Management

School of Geography, Earth and Environmental Sciences, University of

Birmingham Edgbaston, Birmingham B15 2TT, UK

[2] State Environmental Protection Key Laboratory of Urban Ambient Air

Particulate Matter Pollution Prevention and Control, College of Environmental

Science and Engineering

Nankai University, Tianjin 300350, China

18
19
† Also at: Department of Environmental Sciences / Centre of Excellence in Environmental
Studies, King Abdulaziz University, PO Box 80203, Jeddah, 21589, Saudi Arabia.

Corresponding authors:
E-mail: r.m.harrison@bham.ac.uk (Roy M. Harrison)



**ABSTRACT**
Three groups of aliphatic carbonyl compounds, the n-alkanals (C8-C20), n-alkan-2-ones (C8-C26)
and n-alkan-3-ones (C8-C19) were measured in air samples collected in London from January-April
2017. Four sites were sampled including two roof-top background sites, one ground-level urban
background site and a street canyon location on Marylebone Road in central London. The n-alkanals
showed the highest concentrations followed by the n-alkan-2-ones and the n-alkan-3-ones, the latter
having appreciably lower concentrations. It seems likely that all compound groups have both primary
and secondary sources and these are considered in the light of published laboratory work on the
oxidation products of high molecular weight n-alkanes. All compound groups show relatively low
correlation with black carbon and NOx in the background air of London, but in street canyon air
heavily impacted by vehicle emissions, stronger correlations emerge especially for the n-alkanals. It
appears that vehicle exhaust is likely to be a major contributor for concentrations of the n-alkanals
whereas it is a much smaller contributor to the n-alkan-2-ones and n-alkan-3-ones. Other primary
sources such as cooking may be significant but were not evaluated. It seems likely that there is also
a significant contribution from photo-oxidation of n-alkanes and this would be consistent with the
much higher abundance of the n-alkan-2-ones relative to the n-alkan-3-ones if the formation
mechanism were to be through oxidation of condensed phase alkanes. Vapour-particle partitioning
fitted the Pankow model well for the n-alkan-2-ones but less well for the other compound groups,
although somewhat stronger relationships were seen at the Marylebone Road site than at the
background sites.
**Keywords:** Carbonyl compounds; n-alkanals; n-alkan-2-ones; n-alkan-3-ones; organic aerosol;
partitioning;



**1.    INTRODUCTION**
Carbonyl compounds are classified as polar organic compounds, constituting a portion of the
oxygenated organic compounds in atmospheric particulate matter (PM). Aliphatic carbonyl
compounds are directly emitted into the atmosphere from primary biogenic and anthropogenic
sources (Schauer et al., 2001, 2002a, b), as well as being secondary products of atmospheric
oxidation of hydrocarbons (Chacon-Madrid et al., 2010; Zhang et al., 2015; Han et al., 2016).

The most abundant atmospheric carbonyls are methanal (formaldehyde) and ethanal (acetaldehyde),
and many studies have described their emission sources and chemical formation in urban and rural
samples (Duan et al., 2016). Long-chain aliphatic carbonyl compounds have been identified in PM
and reported in few published papers (Gogou et al., 1996; Andreou and Rapsomanikis, 2009), and
these compounds are considered to be formed from atmospheric oxidation processes affecting
biogenic emissions of alkanes. Anthropogenic activity is also considered to be a significant
contributor to the aliphatic carbonyls. Appreciable concentrations of aliphatic carbonyl compounds
have been identified in emissions from road vehicles (Schauer et al., 1999), coal combustion (Oros
and Simoneit, 2000), wood burning (Rogge et al., 1998) and cooking processes (Zhao et al., 2007b,
a), spanning a wide range of molecular weights. Furthermore, chamber studies (Chacon-Madrid and
Donahue, 2011; Algrim and Ziemann, 2016) have demonstrated that the aliphatic carbonyl
compounds are very important precursors of secondary organic aerosol (SOA) when they react with
OH radicals in the presence of $NO_x$.



The oxidation of n-alkanes by hydroxyl radical is considered to be an important source of carbonyl
compounds. It was believed that the n-alkanals with carbon atoms numbering less than 20 indicate
oxidation of alkanes, whereas the higher compounds were usually considered to be of direct biogenic
origin (Rogge et al., 1998). The homologues and isomers of n-alkanals and n-alkanones have been
identified as OH oxidation products of n-alkanes in many chamber and flow tube studies (Zhang et
al., 2015; Schilling Fahnestock et al., 2015; Ruehl et al., 2013). The commonly accepted oxidation
pathways of n-alkanes generally divide into functionalization and fragmentation. Functionalization
occurs when an oxygenated functional group ($-ONO_2$, $-OH$, $-C=O$, $-C(O)O-$ and $-OOH$) is added
to a molecule, leaving the carbon skeleton intact. Alternatively, fragmentation involves C−C bond
cleavage and produces two oxidation products with smaller carbon numbers than the reactant. The
chamber studies of dodecane oxidation have identified 1-undecanal, hexan-3-one, octan-3-one,
heptan-2-one, nonan-2-one and decan-2-one as OH oxidation products (Schilling Fahnestock et al.,
2015; Yee et al., 2012).

In London, with a high population density and a large number of diesel engine vehicles, the aliphatic
hydrocarbons constitute an important fraction of ambient aerosols. Anthropogenic activities and
secondary formation favour the emission and production of carbonyl compounds within the city. The
objectives of the present study were the identification and quantification of aliphatic carbonyl
compounds in particle and vapour samples collected in London from January to April 2017. This
work has aided an understanding of the concentrations and secondary formation of carbonyls in the
London atmosphere. Spatial and temporal variations of the studied carbonyl compounds were
assessed and used to infer sources. One of the main objectives was to provide gas/particle partitioning



coefficients of identified carbonyls under realistic conditions. Diagnostic criteria were used to
estimate the sources of identifiable atmospheric carbonyl compounds. Additionally, for the first time,
concentrations of particulate and gaseous n-alkan-3-ones are reported.

**2.      MATERIALS AND METHODS**
**2.1      Sampling Method and Site Characteristics**
Three sampling campaigns were carried out between 23 January and 18 April 2017 at four sampling
sites (Figure 1) in London. The first campaign used two sampling sites, one located on the roof of a
building (15 m above ground) of the Regent's University (51°31′N, -0°9′W), hereafter referred to as
RU, sampled from 23 January 2017 to 19 February 2017, the other located on the roof (20 m above
ground) of a building which belongs to the University of Westminster on the southern side of
Marylebone Road (hereafter referred to as WM), sampled from 24 January 2017 to 20 February 2017.
The third sampling site was located at ground level at Eltham (51°27′N, 0°4′E), hereafter referred to
as EL, sampled from 23 February 2017 to 21 March 2017, which is located in suburban south London,
and the fourth sampling site was located at ground level on the southern side of Marylebone Road
(51°31′N, -0°9′W), hereafter referred to as MR, sampled from 22 March 2017 to 18 April 2017.
Marylebone Road is in London's commercial centre, and is an important thoroughfare carrying 80-
90,000 vehicles per day through central London. The Regent's University site is within Regent's
Park to the north of Marylebone Road. The Eltham site is in a typical residential neighbourhood, 22
km from the MR site. Earlier work at the Marylebone Road and a separate Regent's Park site is
described by Harrison et al. (2012).





The particle samples were collected on polypropylene backed PTFE filters (47 mm, Whatman) which
preceded stainless steel sorbent tubes packed with 1cm quartz wool, 300 mg Carbograph 2TD 40/60
(Markes International, Llantrisant, UK) and sealed with stainless-steel caps before and after sampling.
Sampling took place for sequential 24-hour periods at a flow rate of 1.5 L min$^{-1}$ using an in-house
developed automated sampler. Field blank filters and adsorption tubes were prepared for each site,
and recovery efficiencies were evaluated. After the sampling, each filter was placed in a clean sealed
petri dish, wrapped in aluminium foil and stored in the freezer at -18ºC prior to analysis. Black carbon
(BC) was simultaneously monitored during the sampling period at RU and WM sites using an
aethalometer (Model AE22, Magee Science). Measurements of BC and $NO_x$ at MR and $NO_x$ at EL
were provided by the national network sites of Marylebone Road, and Eltham (https://uk-
air.defra.gov.uk/).

## 2.2    Analytical Instrumentation

The particle samples were analyzed using a 2D gas chromatograph (GC, 7890A, Agilent
Technologies, Wilmington, DE, USA) equipped with a Zoex ZX2 cryogenic modulator (Houston,
TX, USA). The first dimension was equipped with a SGE DBX5, non-polar capillary column (30.0
m, 0.25 mm ID, 0.25 mm – 5.00% phenyl polysilphenylene-siloxane), and the second-dimension
column equipped with a SGE DBX50 (4.00 m, 0.10 mm ID, 0.10 mm – 50.0% phenyl
polysilphenylene-siloxane). The GC × GC was interfaced with a Bench-ToF-Select, time-of-flight
mass spectrometer (ToF-MS, Markes International, Llantrisant, UK). The acquisition speed was 50.0
Hz with a mass resolution of >1200 fwhm at 70.0 eV and the mass range was 35.0 to 600 m/z. All
data produced were processed using GC Image v2.5 (Zoex Corporation, Houston, US).





### 2.3    Analysis of Samples

Standards used in these experiments included 19 alkanes, $C_8$ to $C_{26}$ (Sigma-Aldrich, UK, purity >99.2%); 12 n-aldehydes, $C_8$ to $C_{13}$ (Sigma-Aldrich, UK, purity ≥95.0%), $C_{14}$ to $C_{18}$ (Tokyo Chemical Industry UK Ltd, purity >95.0%); and 10 2-ketones, $C_8$ to $C_{13}$ and $C_{15}$ to $C_{18}$ (Sigma-Aldrich, UK, purity ≥98.0%) and $C_{14}$ (Tokyo Chemical Industry UK Ltd, purity 97.0%).

The filters were spiked with 30.0 µL of 30.0 µg $mL^{-1}$ deuterated internal standards (dodecane-$d_{26}$, pentadecane-$d_{32}$, eicosane-$d_{42}$, pentacosane-$d_{52}$, triacontane-$d_{62}$, butylbenzene-$d_{14}$, nonylbenzene-2,3,4,5,6-$d_5$, biphenyl-$d_{10}$, p-terphenyl-$d_{14}$; Sigma-Aldrich, UK) for quantification and then immersed in dichloromethane (DCM), and ultra-sonicated for 20.0 min at 20.0℃. The extract was filtered using a clean glass pipette column packed with glass wool and anhydrous $Na_2SO_4$, and concentrated to 50.0 µL under a gentle flow of nitrogen for analysis using GC × GC-ToF-MS. 1 µL of the extracted sample was injected in a split ratio 100:1 at 300℃. The initial temperature of the primary oven (80.0℃) was held for 2.0 min and then increased at 2.0 ℃ $min^{-1}$ to 210℃, followed by 1.5 ℃ $min^{-1}$ to 325 ℃. The initial temperature of the secondary oven (120℃) was held for 2.0 min and then increased at 3.0℃ $min^{-1}$ to 200℃, followed by 2.00℃ $min^{-1}$ to 300℃ and a final increase of 1.0℃ $min^{-1}$ to 330 ℃ to ensure all species passed through the column. The transfer line temperature was 330 ℃ and the ion source temperature was 280℃. Helium was used as the carrier gas at a constant flow rate of 1.0 mL $min^{-1}$. Further details of the instrumentation and data processing methods is given by Alam et al. (2016a,b).





The sorbent tubes were analyzed by an injection port thermal desorption unit (Unity 2, Markes
International, Llantrisant, UK) and subsequently analyzed using GC × GC-ToF-MS. Briefly, the
tubes were spiked with 1 ng of deuterated internal standard for quantification and desorbed onto the
cold trap at 350ºC for 15.0 min (trap held at 20.0ºC). The trap was then purged onto the column in
a split ratio of 100:1 at 350ºC and held for 4.0 min. The initial temperature of the primary oven
(90.0ºC) was held for 2.0 min and then increased to 2.0ºC min$^{-1}$ to 240ºC, followed by 3.0ºC min$^{-1}$
to 310ºC and held for 5.0 min. The initial temperature of the secondary oven (40.0ºC) was held for
2.0 min and then increased at 3.0ºC min$^{-1}$ to 250ºC, followed by an increase of 1.5ºC min$^{-1}$ to 315ºC
and held for 5.0 min. Helium was used as carrier gas for the thermally desorbed organic compounds,
with a gas flow rate of 1.0 mL min$^{-1}$.

*Qualitative analysis*
Compound identification was based on the GC×GC-TOFMS spectra library, NIST mass spectral
library and in conjunction with authentic standards. Compounds within the homologous series for
which standards were not available were identified by comparing their retention time interval between
their homologues, and by comparison of mass spectra to the standards for similar compounds within
the series, by comparison to the NIST mass spectral library and by the analysis of fragmentation
patterns.

*Quantitative analysis*
An internal standard solution (including dodecane-d$_{26}$, pentadecane-d$_{32}$, eicosane-d$_{42}$, pentacosane-
d$_{52}$, triacontane-d$_{62}$, nonylbenzene-2,3,4,5,6-d$_5$, butylbenzene-d$_{14}$, biphenyl-d$_{10}$, p-terphenyl-d$_{14}$)



(Sigma-Aldrich, UK) was added to the samples to extract prior to instrumental analysis. Five internal
standards (pentadecane-$d_{32}$, eicosane-$d_{42}$, pentacosane-$d_{52}$, triacontane-$d_{62}$, nonylbenzene-2,3,4,5,6-
$d_5$) were used in the calculation of carbonyl compound concentrations.

The quantification for alkanes, aldehydes and 2-ketones was performed by the linear regression
method using seven-point calibration curves (0.05, 0.10, 0.25, 0.50, 1.00, 2.00, 3.00 ng $\mu L^{-1}$)
established between the authentic standards/internal standard concentration ratios and the
corresponding peak area ratios. The calibration curves for all target compounds were highly linear
($r^2 > 0.99$, from 0.990 to 0.997), demonstrating the consistency and reproducibility of this method.
Limits of detection for individual compounds were typically in the range 0.04–0.12 ng $m^{-3}$. 3-ketones
were quantified using the calibration curves for 2-ketones. This applicability of quantification of
individual compounds using isomers of the same compound functionality (which have authentic
standards) has been discussed elsewhere and has a reported uncertainty of 24% (Alam et al., 2018).

Alkan-2-ones and alkan-3-ones were not well separated by the chromatography. These were separated
manually using the peak cutting tool, attributing fragments at m/z 58 and 71 to 2-ketones and m/z 72
and 85 to 3-ketones. The calibration for 2-ketones was applied to quantification of the 3-ketones.

Field and laboratory blanks were routinely analysed to evaluate analytical bias and precision. Blank
levels of individual analytes were normally very low and in most cases not detectable. Recovery
efficiencies were determined by analyzing the blank samples spiked with standard compounds. Mean



recoveries ranged between 78.0 and 102%. All quantities reported here have been corrected according
to their recovery efficiencies.

**3.     RESULTS AND DISCUSSION**
**3.1     Mass Concentration of Particle-Bound Carbonyl Compounds**
Fig. 2 shows the average total concentrations of particle-bound 1-alkanals, n-alkan-2-ones, and n-
alkan-3-ones from January to April at four measurement sites, and the particle and gaseous phase
concentrations are detailed in the Table S1 (Supporting Information). Total n-alkanals was defined
as the sum of particle-bound n-alkanals ranging from $C_8$ to $C_{20}$. The particulate n-alkanals at the MR
site accounted for 75.2% of the measured particle carbonyls with the average total concentration of
682 ng m$^{-3}$, and concentrations at the other sites were 167 ng m$^{-3}$ at EL, 117 ng m$^{-3}$ at WM and 82.6
ng m$^{-3}$ at RU, accounting for 57.0%, 57.9% and 56.3% of the measured particulate carbonyls,
respectively. The n-alkanals identified in this study differed in some aspects from those previously
reported in samples collected from Crete (Gogou et al., 1996) and Athens (Andreou and
Rapsomanikis, 2009) in Greece. The n-alkanals from London presented narrower ranges of carbon
numbers and a higher concentration than rural and urban samples from Crete. The concentrations of
n-alkanal homologues ($C_8$-$C_{20}$) ranged from 5.50 to 141 ng m$^{-3}$ (average 52.0 ng m$^{-3}$) at MR which
were far higher than 1.48-28.6 ng m$^{-3}$ (average 6.44 ng m$^{-3}$) at RU, 1.42-50.3 ng m$^{-3}$ (average 9.03
ng m$^{-3}$) at WM and 3.29-53.0 ng m$^{-3}$ (average 13.0 ng m$^{-3}$) at EL (Table S1), unlike Crete where the
concentrations were 0.9-3.7 ng m$^{-3}$ in rural ($C_{15}$-$C_{30}$) and 5.4-6.7 ng m$^{-3}$ in urban ($C_9$-$C_{22}$) samples,
and the average concentration of all four sites was much higher than the 0.91 ng m$^{-3}$ measured in
Athens (Andreou and Rapsomanikis, 2009) ($C_{13}$-$C_{20}$).



As part of the CARBOSOL project (Oliveira et al., 2007), air samples were collected in summer and
winter at six rural sites across Europe. The particulate n-alkanals ranged from $C_{11}$ to $C_{30}$ with average
total concentrations between 1.0 ng m$^{-3}$ and 19.0 ng m$^{-3}$, with higher concentrations in summer than
winter at all but one site. These concentrations fall well below those measured in the present study,
although the range of compounds differed. Maximum concentrations at all sites were in
compounds >$C_{22}$ indicating a source from leaf surface abrasion products and biomass burning. This
far exceeds the $C_{max}$ values seen in the particulate fraction at our sites.

The n-alkan-2-one homologues measured in London ranged from $C_8$ to $C_{26}$, and the average total
particulate fraction concentration was 58.5 ng m$^{-3}$ at RU, 75.1 ng m$^{-3}$ at WM, 112 ng m$^{-3}$ at EL and
186 ng m$^{-3}$ at MR, approximately accounting for 39.9% (RU), 37.0% (WM), 38.1% (EL) and 20.5%
(MR) of the total particulate carbonyls, respectively (Fig. 2). The published data from Greece
indicated that the concentrations of n-alkan-2-ones were independent of the seasons, and an average
of 5.40 ng m$^{-3}$ ($C_{13}$-$C_{29}$) was measured in August and 5.44 ng m$^{-3}$ in March at Athinas St, but 12.88
ng m$^{-3}$ was measured in March at the elevated (20 m) AEDA site in Athens (Gogou et al. (1996).
Concentrations in Crete for alkan-2-ones ($C_{10}$-$C_{31}$) were 0.4-2.1 ng m$^{-3}$ at the rural site and 1.9-2.6
ng m$^{-3}$ at the urban site (Andreou and Rapsomanikis, 2009).

The CARBOSOL project also determined concentrations of n-alkan-2-ones, between $C_{14}$ and $C_{31}$
with a $C_{max}$ at $C_{28}$ or $C_{29}$ at all but one site. Average concentrations ranged from 0.15 ng m$^{-3}$ ($C_{17-29}$)
to 3.35 ($C_{14}$-$C_{31}$), very much below the concentrations at our London sampling site. Cheng et al.
(2006) measured concentrations of n-alkan-2-ones in the Lower Fraser Valley, Canada, in PM$_{2.5}$.

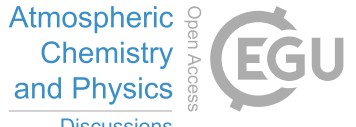

Samples collected in a road tunnel showed the highest concentrations, total 1.8-12.6 ng m$^{-3}$ for C$_{10}$-
C$_{31}$, and were higher in daytime than nighttime. Concentrations at a forest site were 1.1-7.2 ng m$^{-3}$
without a diurnal pattern. Values of C$_{max}$ ranged from C$_{16-17}$ at the road tunnel to C$_{27}$ (secondary
maximum) at the forest site. Values of CPI averaged across sites from 1.00 to 1.34, giving little
evidence for a substantial biogenic input from higher plant waxes.

Atmospheric concentrations of long-chain n-alkan-3-ones have not previously been reported in the
literature. The n-alkan-2-one and n-alkan-3-one homologues with few carbon atoms are believed
mainly to originate as the fragmental products of n-alkanes (Yee et al., 2012; Schilling Fahnestock
et al., 2015), whereas the higher compounds are mainly generated from functional pathways (Zhang
et al., 2015; Ruehl et al., 2013). The n-alkan-3-one homologues identified in the samples ranged
from C$_8$ to C$_{19}$, and the average of individual compound concentrations was 0.52 ng m$^{-3}$ at RU, 0.94
ng m$^{-3}$ at WM, 1.37 ng m$^{-3}$ at EL and 3.34 ng m$^{-3}$ at MR. The concentrations of n-alkan-3-ones at
the four sites were lower than the n-alkanals and n-alkan-2-ones, and MR had the highest average
total mass concentrations 39.4 ng m$^{-3}$, followed by 14.3 ng m$^{-3}$ at EL, 10.4 ng m$^{-3}$ at WM and 5.65
ng m$^{-3}$ at RU, respectively.

Recently published studies have found that the isomeric distribution of first-generation oxidation
products of n-alkanes depends strongly upon whether the reaction occurs in the gas phase or at the
particle surface (Kwok and Atkinson, 1995; Ruehl et al., 2013). The homogeneous gas-phase
oxidation occurs fast, and H-abstraction by OH radicals occurs at all carbon sites. The fractions of
the OH radical reaction by H atom abstraction from n-decane at the 1-, 2-, 3-, 4- and 5-positions are





3.10%, 20.7%, 25.4%, 25.4%, and 25.4%, respectively, and the products from homogeneous reaction
were generally in accord with structure-reactivity relationship (SRR) predictions (Kwok and
Atkinson, 1995; Aschmann et al., 2001). Reaction of particulate n-alkanes is dominated by
heterogeneous reactions with OH, and the H-abstraction occurs preferentially at the 2-position of the
carbon chain (Zhang et al., 2015; Ruehl et al., 2013). The n-alkanes diffuse from the inner particle to
the surface, where the OH will quickly attack the H atom of 1 and 2 position carbons. The intermediate
products at the 2-position are relatively more stable than at the 1-position, and the products are
dominated by oxidation of the 2-position. The isomeric carbonyls formed via OH-initiated
heterogeneous reactions of n-octacosane ($C_{28}$) exhibit a pronounced preference at the 2-position of
the molecule chain[18]. The n-octacosan-2-ones have the highest relative yield (1.00), followed by n-
octacosan-3-ones (0.50), while other isomeric carbonyl yields were lower than 0.20. The same results
were found in the subsequent chamber studies of n-alkanes (Zhang et al., 2015) ($C_{20}$, $C_{22}$, $C_{24}$) but
not $C_{18}$. The main reason was that OH oxidation of $C_{18}$ was dominated by the homogeneous reaction
as a large fraction of $C_{18}$ evaporated into the gas phase.

During the field experiment, the 1-alkanal homologues were abundant in all samples, and this could
be explained by a strong impact of anthropogenic activities. Thus, the n-alkanals are considered to
arise mainly from primary emission sources. Furthermore, the particulate form of the n-alkane
homologues ($C_{14}$-$C_{36}$) identified in the samples ranged from 50-100% in contrast to the low MW n-
alkanes ($C_{11}$-$C_{13}$). The H-abstraction by OH radicals may therefore have been dominated by
heterogeneous reactions generating the higher concentrations of n-alkan-2-ones than n-alkan-3-ones
that were found in all samples. The ratio of n-alkan-2-ones/n-alkan-3-ones ($C_{11}$-$C_{18}$) with the same





carbon atom number ranged from 2.35-11.3 at four measurement sites. Surprisingly, although the n-
alkane ($C_{11}$-$C_{13}$) oxidation was expected to be dominated by homogeneous reactions, the n-alkan-2-
one/n-alkan-3-one ratios were still greater than 2.00. The probable reason was that the lower
molecular weight n-alkan-2-ones were significantly impacted by primary emission sources. Another
likely reason is that the n-alkan-2-one and n-alkan-3-one homologues with lower carbon atom
numbers originated in part from the fragmental products of higher n-alkanes (Yee et al., 2012;
Schilling Fahnestock et al., 2015).

The ratios of n-alkan-2-ones/n-alkanes, n-alkan-3-ones/n-alkanes (with same carbon numbers) were
calculated and are reported in Table S2. The n-alkan-3-ones with carbon numbers higher than $C_{20}$
were not identified in the samples, indicating that both the homogeneous and heterogeneous reactions
of higher molecular weight n-alkanes were slow, the former probably due to the low vapour phase
presence of n-alkanes. The ratios of n-alkan-3-ones/n-alkanes at four measurement sites gradually
increased from $C_{11,}$ and then decreased from $C_{17}$, while higher ratios of n-alkan-2-ones/n-alkanes were
observed in the range from $C_{17}$ to $C_{22}$, probably indicating a shift from homogeneous reactions to
heterogeneous reactions with the increase of carbon numbers. The low ratios of n-alkan-2-ones/n-
alkanes with carbon numbers from $C_{23}$ to $C_{26}$ were attributed to the low diffusion rate from the inner
particle to the surface with the increasing carbon number of n-alkanes, even though heterogeneous
reactions were the dominant pathway.







## 3.2 Temporal and Spatial Variations


The study of temporal and spatial variations of air pollutants can provide valuable information about
their sources and atmospheric processing. The time series of particle-bound n-alkanals, n-alkan-2-
ones, and n-alkan-3-ones are plotted in Fig. 3. It is clear that the concentrations of n-alkanals varied
substantially with date, and were always higher than n-alkanones at four sites. It is also clear from
Figure 2 that concentrations were broadly similar at the background sites, RU, WM and EL, but are
elevated, especially for the n-alkanals, at MR. This is strongly indicative of a road traffic source.

## 3.3 Sources of Carbonyl Compounds


### 3.3.1 Homologue distribution and carbon preference index (CPI)


Fig. 4 shows the average concentrations, and molecular distributions of particle-bound carbonyl
compounds at the four sites. The values of carbon preference index (CPI) were calculated to estimate
the origin of carbonyl compounds, according to Bray and Evans (1961):

$$\text{CPI} = \frac{1}{2}\left(\frac{\sum_4^m C_{2i+1}}{\sum_4^m C_{2i}} + \frac{\sum_4^m C_{2i+1}}{\sum_5^{m+1} C_{2i}}\right)$$
For n-alkanals and n-alkan-3-ones (m=9): $\text{CPI} = \frac{1}{2}\left(\frac{\sum odd(C_9-C_{19})}{\sum even(C_8-C_{18})} + \frac{\sum odd(C_9-C_{19})}{\sum even(C_{10}-C_{20})}\right)$
For n-alkan-2-ones (m=12): $\text{CPI} = \frac{1}{2}\left(\frac{\sum odd(C_9-C_{25})}{\sum even(C_8-C_{24})} + \frac{\sum odd(C_9-C_{25})}{\sum even(C_{10}-C_{26})}\right)$

where $i$ takes values between 4 and $m$, and 5 and $m$ as in the equation, and
$m = 9$ for n-alkanal and n-alkan-3-ones
$m = 12$ for n-alkan-2-ones



The carbon maximum number ($C_{max}$) was used to evaluate the relative contribution of the source and
exhibit the homologue distribution of highest concentration. Table. 1 presents the CPI and $C_{max}$ of
particle-bound carbonyl compounds calculated in the current and other studies.
According to the low CPI (0.41-1.07) at four sites, the n-alkanal homologues with carbon number
from $C_8$ to $C_{20}$ mainly originate from anthropogenic emissions or OH oxidation of anthropogenic
hydrocarbons. The particle-bound n-alkanals exhibited a similar distribution of carbon number from
January to April at four sites, and they had the same $C_{max}$ at $C_8$ with concentration 28.6 ng m$^{-3}$ at
RU, 50.3 ng m$^{-3}$ at WM, 53.0 ng m$^{-3}$ at EL and 141 ng m$^{-3}$ at MR, respectively. This compound may
be a fragmentation product, oxidation product or primary emission. In addition, the distribution of
n-alkanals had a second concentration peak at $C_{15}$ (MR) and $C_{18}$ (RU, WM, and EL). The $C_{18}$
compound was observed accounting for the highest percentage of the total mass of n-alkanals in
some rural aerosol samples (Gogou et al., 1996) in Crete. Andreou and Rapsomanikis reported the
$C_{max}$ as $C_{15}$ or $C_{17}$ in Athens (Andreou and Rapsomanikis, 2009) and attributed this to the oxidation
of n-alkanes. However, a $C_{max}$ at $C_{26}$ or $C_{28}$ in urban Crete (Gogou et al., 1996) was observed,
suggestive of biogenic input. The homologue distribution and CPI of n-alkanals in this study differed
from those previous reports, and demonstrated weak biogenic input and a strong impact of
anthropogenic activities in the London samples.

In this study, n-alkan-2-ones have similar homologue distributions and $C_{max}$ ($C_{19}$ or $C_{20}$) (Table 2)
at RU, WM and EL sites, and the total concentration from $C_{16}$ to $C_{23}$ accounts for 76.0%, 76.1% and
68.0% of $\sum$n-alkan-2-ones, respectively. The CPI values for n-alkan-2-ones ranged from 0.57 to
1.23 at the RU, MR and WM sites and were not indicative of biogenic input, and were considered





to mainly originate from anthropogenic activities and OH oxidation of anthropogenic n-alkanes.
At EL, the CPI of 1.57 is probably indicative of a biogenic contribution in suburban south London.
A difference was observed at the MR site, the n-alkan-2-ones with carbon atoms numbering from
$C_{12}$ to $C_{18}$ accounting for 72.0% of $\sum$n-alkan-2-ones, with the $C_{max}$ being at $C_{16}$. The $C_{max}$ of n-alkan-
3-ones was at $C_{16}$ at the MR site, at EL, $C_{max} = C_{16}$, WM, $C_{max} = C_{17}$ and at RU, $C_{max} = C_{17}$,
respectively.

### 3.3.2    The ratios of n-alkanes/n-alkanals

Diesel engine emission studies have been conducted previously in our group; details of the engine set
up and exhaust sampling system are given elsewhere (Alam et al., 2016b). Briefly, the steady-state
diesel engine operating conditions were at a load of 5.90 bar mean effective pressure (BMEP) and a
speed of 1800 revolutions per minute (RPM), and samples (n=14) were collected both before a diesel
oxidation catalyst (DOC) and after a diesel particulate filter (DPF). The n-alkanes ($C_{12}$ - $C_{37}$) and 1-
alkanals ($C_9$ - $C_{18}$) were quantified in the particle samples, while n-alkanones were not identified
because their concentrations were lower than the limits of (detection 0.01–0.15 ng m$^{-3}$). The emission
concentrations of n-alkanals ranged from 7.10 to 53.2 µg m$^{-3}$ (before DOC) and 1.20 to 11.5 µg m$^{-3}$
(after DPF), respectively, and the ratios of alkanes/alkanals ($C_{12}$-$C_{18}$) with the same carbon atom
numbers ranged from 0.15 to 0.23 (before DOC) and 0.52 to 7.60 (after DPF). The n-alkane/n-alkanal
($C_{12}$-$C_{18}$) ratio at MR ranged from 0.92 to 5.03, while average ratios of 27.6 (RU), 22.1 (WM) and
15.1 (EL) were obtained, respectively. The similarity of the n-alkanes/n-alkanal ratio between MR
and the engine studies (after DPF) strongly suggests that diesel vehicle emissions were the main
source of 1-alkanals at MR.



The emission factors of total alkanes from diesel engines are reported to be 7 times greater than
gasoline engines (Perrone et al., 2014), with n-alkanals with carbon atoms numbering lower than $C_{11}$
being quantified in the exhaust from gasoline engines (Schauer et al., 2002b; Gentner et al., 2013).
The n-alkane/n-alkanal ($C_8$-$C_{10}$) ratio with the same carbon numbers ranged from 5.60 to 14.3,
suggesting that gasoline combustion may be another source of atmospheric n-alkanals.

Studies of n-alkanals showed that aldehydes have high reactivity when the OH radical attacks the
aldehyde moiety (Chacon-Madrid and Donahue, 2011; Chacon-Madrid et al., 2010), and the rate
constants are more than 3 times those of n-alkanes with the same carbon number. The mechanism
and rate constants of H-abstraction by OH detailed in the Master Chemical Mechanism (MCM,
v3.3.1), were obtained via http://mcm.leeds.ac.uk/MCM, and used in the evaluation of our data.

### 3.3.3    Correlation analysis

Insights into the sources of carbonyls can be gained from correlation analysis with black carbon (BC)
and $NO_x$. This has the advantage of comparing relative concentrations of pollutants, rather than
absolute concentrations. The latter are strongly affected by weather conditions, making inter-site
comparisons difficult when sampling did not occur simultaneously. In London, both black carbon
and $NO_x$ arise very substantially from diesel vehicle emissions (Liu et al., 2014; Harrison et al.,
2012; Harrison and Beddows, 2017), and hence these are good measures of road traffic activity. The
concentrations of BC were simultaneously determined by the online instruments during the sampling
periods, with the average concentrations of 1.34, 1.94 and 3.58 µg m$^{-3}$ at the RU, WM and MR sites,
respectively. The data for $NO_x$ were provided by the national network sites, with the average





concentrations of 23.4 and 202 µg m$^{-3}$ at the EL and MR sites, respectively. At the MR site, the
concentrations of BC and NO$_x$ averaged 5.00 µg m$^{-3}$ and 281 µg m$^{-3}$ when southerly winds were
dominant compared to 2.60 and 128 µg m$^{-3}$ for northerly winds. All correlations were carried out
with the sum of particle and vapour phases for the carbonyl compounds, and strong ($r^2 = 0.87$) and
weak ($r^2 = 0.12$) correlations between BC and NO$_x$ were obtained when the southerly and northerly
winds were prevalent at MR, respectively. Marylebone Road is a street canyon site where a vortex
circulation is established by the wind. The effect is that on northerly wind sectors the sampling site
on the southern side of the road samples near-background air, while on southerly wind sectors, the
traffic pollution is carried to the sampling site, leading to elevated pollution levels affected heavily
by the traffic emissions. The strong correlation between BC and NO$_x$ with southerly wind sectors is
a reflection of their emission from road traffic. In addition, the correlations between n-alkanals (C$_8$-
C$_{20}$) and BC, and between n-alkanals (C$_8$-C$_{20}$) and NO$_x$ were calculated to assess the contribution of
vehicular emission (Table S3). The results showed that the correlations ($r^2$) between n-alkanals and
BC gradually decreased from 0.61 (C$_9$) to 0.34 (C$_{20}$) at MR when the southerly winds were prevalent,
indicating that the distribution of n-alkanals, and especially the lower MW compounds, was
significantly impacted by the vehicular exhaust emissions. The average correlations at MR
(southerly winds) between n-alkanals and BC, and between n-alkanals and NO$_x$ were $r^2 = 0.47$ and
$r^2 = 0.32$, respectively. These moderate correlations demonstrated that the vehicular emissions were
a substantial source of n-alkanals at MR, and result in the high background concentrations of n-
alkanals in London. The other probable sources of n-alkanals include cooking emissions, wood
burning, photooxidation of hydrocarbons and industrial emissions. Poorer correlations between n-
alkanals and BC (average $r^2 = 0.15$), and between n-alkanals and NO$_x$ (average $r^2 = 0.15$) were



observed at MR in the north London background air sampled when northerly winds were prevalent.
There were very weak correlations (average $r^2 < 0.10$) between n-alkanals and BC, and between n-
alkanals and $NO_x$ at the RU, WM and EL sites, which may be attributable to the high chemical
reactivity of n-alkanals. High concentrations of furanones (γ-lactones) are generated via the photo-
oxidation reaction of n-alkanals (Alves et al., 2001), and the total concentrations (particle and gas)
were up to 376, 279, 347 and 318 ng m$^{-3}$ at RU, WM, WL, and MR, respectively for the sum of
furanone homologues (from 5-propyldihydro-2(3H)-furanone to 5-tetradecyldihydro-2(3H)-
furanone).

The relationships ($r^2$ values) between BC and $NO_x$ and the n-alkan-2-ones were low at all sites, but
notably higher with southerly winds at MR (average $r^2 = 0.33$ and 0.35 for BC and $NO_x$ respectively)
than for northerly winds ($r^2 = 0.16$ and 0.03 respectively). This is strongly suggestive of a
contribution from vehicle exhaust to n-alkan-2-one concentrations, but smaller than that for n-
alkanals. In the case of the n-alkan-3-ones, correlations averaged $r^2 = 0.25$ with BC and $r^2 = 0.21$ for
$NO_x$ in southerly winds, compared to $r^2 = 0.08$ and $r^2 = 0.05$ respectively for northerly winds. This
is also suggestive of a small, but not negligible contribution of vehicle emissions to n-alkan-3-ones.
The very low correlations observed in background air for both n-alkan-2-ones and n-alkan-3-ones
with BC and $NO_x$ are suggestive of the importance of non-traffic sources, probably including
oxidation of n-alkanes. The considerable predominance for n-alkan-2-one over n-alkan-3-one
concentrations may be indicative of a formation pathway from oxidation of condensed phase n-
alkanes, but this is speculative as primary emissions may be dominant.



## 3.4      The Partition Between Particle and Gas Phase
The partitioning coefficient $K_p$ between particles and vapour was calculated in this study according
to the following equation defined by Pankow (1994):

$$K_p = \frac{c_p}{c_g * TSP}$$

Where, $C_p$ and $C_g$ ($\mu g\ m^{-3}$) are the concentration of the compounds in the particulate phase and
gaseous phase, respectively. TSP is the concentration of total suspended particulate matter ($\mu g\ m^{-3}$),
which was estimated from the $PM_{10}$ concentration ($PM_{10}/TSP = 0.80$), and daily average $PM_{10}$
concentrations were taken from the national network sites. The partitioning coefficients $K_p$
calculated from our data and the percentages in the particulate form are presented in Table 2. For
the three types of carbonyls, the n-alkanals $>C_{16}$, n-alkan-2-ones $>C_{19}$, and n-alkan-3-ones $> C_{18}$
were assumed to have negligible vapour concentrations, and the partitioning into the particulate
phase gradually increased from $C_8$ to high molecular weight compounds.

Log Kp was regressed against vapour pressure ($VP_T$) for the relevant temperature derived from
UManSysProp (http://umansysprop.seaes.manchester.ac.uk/) according to the following equation:

$$\text{Log } K_p = m \log(VP_T) + b$$

The calculated log $K_p$ versus log ($VP_T$) for the three types of carbonyls was calculated for each day,
and the results appear in the Table S4. Data from four sites were over the temperature range 0.40–
15.3 °C. A good fit to the data for n-alkan-2-ones ($r^2 = 0.54$–$0.94$ at RU, 0.64-0.93 at WM, 0.43-



0.95 EL and 0.45-0.89 at MR) was obtained. It is notable that the fit to the regression equation as
indicated by the $r^2$ value is appreciably higher at the MR site than at the other sites, especially in the
case of the alkan-3-ones. This is not easily explained, except perhaps by an increased particle surface
area at the MR site which may enhance the kinetics of gas-particle exchange, leading to partitioning
which is closer to equilibrium.

**4.    CONCLUSIONS**
Three groups of carbonyl compounds were determined in the particle and gaseous phase in London
and concentrations are reported for n-alkanals ($C_8$-$C_{20}$), n-alkan-2-ones ($C_8$-$C_{26}$) and n-alkan-3-ones
($C_8$-$C_{19}$). The Marylebone Road site has the highest concentration of particle-bound n-alkanals, and
the average total concentration was up to 682 ng m$^{-3}$, followed by 167 ng m$^{-3}$ at EL, 117 ng m$^{-3}$ at
WM and 82.6 ng m$^{-3}$ at RU. The particulate n-alkanals were abundant in all samples at all four
measurement sites, accounting for more than 56.3% of total particle carbonyls. In addition, the
average total particle concentrations of n-alkan-2-ones and n-alkan-3-ones at four measurement sites
were in the range of 58.5-186 ng m$^{-3}$ and 5.65-39.4 ng m$^{-3}$, respectively. Diagnostic criteria,
including molecular distribution, CPI, $C_{max}$, ratios and correlations, were used to assess the sources
and their contributions to carbonyl compounds. The three groups of carbonyls have similar
molecular distributions and $C_{max}$ values at the four measurement sites, and their low CPI values
(0.41-1.57) at the four sites indicate a weak biogenic input during sampling campaigns. Heavily
traffic-influenced air and urban background air were measured at the MR site when southerly and
northerly winds were prevalent respectively; correlations of $r^2 = 0.47$ and $r^2=0.32$ were obtained
between n-alkanals and BC, and between between n-alkanals and NO$_x$, respectively in southerly





winds. Vehicle emissions appear to be an important source of n-alkanals, which is confirmed by the
similar ratios of n-alkanes/n-alkanals measured at MR (0.92-5.03) and in diesel engine exhaust
studies (0.52-7.6), resulting in a high background concentration in London. In addition, the OH-
initiated heterogeneous reactions of n-alkanes appear to be important sources of n-alkanones, even
though weak contributions from vehicular exhaust emissions were suggested by correlation analysis
with BC and $NO_x$ in southerly winds at MR. Anthropogenic primary sources appear to account for
a large proportion of the alkan-2-one and alkan-3-one concentrations measured in London.

In addition, the partitioning coefficients of carbonyls were determined from the relative proportions
of the particle and gaseous phases of individual compounds. The results of field measurements of
partitioning between particle and vapour phases showed generally a better fit at MR than at the other
three sites. The n-alkan-2-ones have a better fit at four sites than the n-alkanals and n-alkan-3-ones,
with $r^2$ = 0.78 (0.54–0.94) at RU, 0.85 (0.64-0.93) at WM, 0.74 (0.43-0.95) EL and 0.70 (0.45-0.89)
at MR, respectively in a regression of log $K_p$ versus the compound vapour pressure.

**ACKNOWLEDGEMENTS**
Primary collection of samples took place during the FASTER project which was funded by the
European Research Council (ERC-2012-AdG, Proposal No. 320821). The authors would also like
to thank the China Scholarship Council (CSC) for support to R.L., and the Natural Environment
Research Council for support under the Air Pollution and Human Health (APHH) programme
(NE/N007190/1).





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



**TABLE LEGENDS**

Table 1.    The carbon preference index (CPI) and $C_{max}$ for n-alkanals, n-alkan-2-ones, and
n-alkan-3-ones in this study and published data.

Table 2.    Percentages of particle phase form and the partitioning coefficient Kp.

**FIGURE LEGENDS**

Figure 1.    Map of the sampling sites. RU-Regents University (15 m above ground); WM-
University of Westminster (20 m above ground); EL-Eltham; MR-Marylebone Road
(south side).

Figure 2.    The average total concentration of particle-bound n-alkanals ($C_8$-$C_{20}$), n-alkan-2-ones
($C_8$-$C_{26}$), and n-alkan-3-ones ($C_8$-$C_{19}$), for each sampling period and site. The error bars
indicate one standard deviation.

Figure 3.    Time series of particle-bound $\sum$1-alkanals, $\sum$n-alkan-2-ones and $\sum$n-alkan-3-ones at
RU, WM, EL, and MR sites.

Figure 4.    The molecular distribution of particle-bound carbonyl compounds at four sites (RU,
WM, EL, and MR).

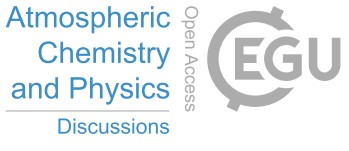

Table 1. The carbon preference index (CPI) and $C_{max}$ for n-alkanals, n-alkan-2-ones, and n-alkan-3-ones in this study and published data.

| Location / Sampling site | Sampling period | n-alkanals | | n-alkan-2-ones | | n-alkan-3-ones | | Reference |
|---|---|---|---|---|---|---|---|---|
| | | CPI | $C_{max}$ | CPI | $C_{max}$ | CPI | $C_{max}$ | |
| RU, surrounded by Regent's Park, 15 m above ground | 23 Jan - 19 Feb | 0.52 | $C_8$ | 1.23 | $C_{19}$ | 1.30 | $C_{17}$ | Present study |
| WM, 20 m above ground | 24 Jan - 20 Feb | 0.41 | $C_8$ | 0.99 | $C_{20}$ | 1.26 | $C_{17}$ | Present study |
| EL, suburb of London | 23 Feb - 21 Mar | 0.71 | $C_8$ | 1.57 | $C_{20}$ | 1.04 | $C_{16}$ | Present study |
| MR, adjacent to Marylebone road | 22 Mar - 18 Apr | 1.07 | $C_8$ | 0.57 | $C_{16}$ | 1.12 | $C_{16}$ | Present study |
| Athens, Athinas St. Urban roadside | August | 1.49 | $C_{15}$, $C_{17}$ | 1.09 | $C_{18}$, $C_{21}$, $C_{19}$ | | | (Andreou and Rapsomanikis, 2009) |
| | March | | | 3.26 | $C_{21}$, $C_{19}$, $C_{20}$ | | | |
| Athens, AEDA, Urban, 20 m above ground | March | | | 2.41 | $C_{19}$, $C_{18}$, $C_{20}$ | | | (Andreou and Rapsomanikis, 2009) |
| Heraklion, Greece Urban 15 m above ground | Spring /summer | 0.80–1.40 | $C_{26}$, $C_{28}$ | 1.30–1.80 | $C_{23}$, $C_{29}$, $C_{31}$ | | | (Gogou et al., 1996) |
| Vancouver, Canada Roadway tunnel | | | | 1.33 | $C_{17}$, $C_{19}$ | | | (Cheng et al., 2006) |
| Aveiro, Portugal Suburban | Summer | | $C_{22}$, $C_{23}$, $C_{26}$ | | $C_{26}$, $C_{28}$, $C_{30}$ | | | |
| | Winter | | | | | | | |
| K-Puszta, Hungary | Summer | | $C_{24}$, $C_{26}$, $C_{28}$ | | $C_{24}$, $C_{26}$, $C_{28}$ | | | (Oliveira et al., 2007) |



Table 2. Percentages of particle phase form and the partitioning coefficient Kp.

| | RU | | | | | | WM | | | | | |
| --- | --- | --- | --- | --- | --- | --- | --- | --- | --- | --- | --- | --- |
| | n-alkanals | | n-alkan-2-ones | | n-alkan-3-ones | | n-alkanals | | n-alkan-2-ones | | n-alkan-3-ones | |
| | % | Kp | % | Kp | % | Kp | % | Kp | % | Kp | % | Kp |
| $C_8$ | 82.9 | 1.16E-04 | 18.4 | 5.37E-06 | 23.9 | 7.47E-06 | 80.2 | 9.09E-05 | 13.3 | 3.43E-06 | 34.1 | 1.16E-05 |
| $C_9$ | 69.2 | 5.37E-05 | 14.5 | 4.03E-06 | 16.6 | 4.74E-06 | 60.5 | 3.43E-05 | 15.6 | 4.16E-06 | 28.7 | 9.05E-06 |
| $C_{10}$ | 75.3 | 7.27E-05 | 13.6 | 3.77E-06 | 7.43 | 1.92E-06 | 82.1 | 1.03E-04 | 14.4 | 3.77E-06 | 23.3 | 6.82E-06 |
| $C_{11}$ | 45.5 | 1.99E-05 | 21.4 | 6.49E-06 | 12.8 | 3.49E-06 | 62.4 | 3.72E-05 | 20.1 | 5.65E-06 | 36.3 | 1.28E-05 |
| $C_{12}$ | 74.8 | 7.08E-05 | 25.0 | 7.96E-06 | 31.3 | 1.09E-05 | 73.7 | 6.29E-05 | 28.8 | 9.07E-06 | 22.7 | 6.60E-06 |
| $C_{13}$ | 82.9 | 1.15E-04 | 61.0 | 3.74E-05 | 35.4 | 1.31E-05 | 82.2 | 1.04E-04 | 48.9 | 2.14E-05 | 62.5 | 3.74E-05 |
| $C_{14}$ | 82.8 | 1.15E-04 | 49.5 | 2.34E-05 | 35.5 | 1.31E-05 | 75.8 | 7.04E-05 | 31.8 | 1.05E-05 | 25.6 | 7.74E-06 |
| $C_{15}$ | 99.5 | 5.01E-03 | 84.1 | 1.26E-04 | 50.5 | 2.44E-05 | 100 | | 85.0 | 1.27E-04 | 68.5 | 4.87E-05 |
| $C_{16}$ | 100 | | 91.4 | 2.53E-04 | 70.3 | 5.64E-05 | 100 | | 89.6 | 1.93E-04 | 91.7 | 2.47E-04 |
| $C_{17}$ | 100 | | 91.5 | 2.55E-04 | 100 | | 100 | | 85.9 | 1.36E-04 | 91.5 | 2.42E-04 |
| $C_{18}$ | 100 | | 94.1 | 3.80E-04 | 100 | | 100 | | 84.8 | 1.26E-04 | 99.4 | 4.02E-03 |
| $C_{19}$ | 100 | | 99.1 | 2.69E-03 | | | 100 | | 100 | | | |
| $C_{20}$ | 100 | | 100 | | | | 100 | | 100 | | | |
| $C_{21}$ | | | 100 | | | | | | 100 | | | |
| $C_{22}$ | | | 100 | | | | | | 100 | | | |
| $C_{23}$ | | | 100 | | | | | | 100 | | | |
| $C_{24}$ | | | 100 | | | | | | 100 | | | |
| $C_{25}$ | | | 100 | | | | | | 100 | | | |
| $C_{26}$ | | | 100 | | | | | | 100 | | | |



| | EI | | | | | | MR | | | | | |
|---|---|---|---|---|---|---|---|---|---|---|---|---|
| | n-alkanals | | n-alkan-2-ones | | n-alkan-3-ones | | n-alkanals | | n-alkan-2-ones | | n-alkan-3-ones | |
| | % | Kp | % | Kp | % | Kp | % | Kp | % | Kp | % | Kp |
| $C_8$ | 92.7 | 6.53E-04 | 24.9 | 1.72E-05 | 31.9 | 2.43E-05 | 90.0 | 2.94E-04 | 28.2 | 1.28E-05 | 43.0 | 2.46E-05 |
| $C_9$ | 92.2 | 6.16E-04 | 38.0 | 3.18E-05 | 44.4 | 4.15E-05 | 89.9 | 2.89E-04 | 27.0 | 1.20E-05 | 39.1 | 2.09E-05 |
| $C_{10}$ | 90.5 | 4.96E-04 | 47.6 | 4.70E-05 | 47.0 | 4.59E-05 | 91.7 | 3.62E-04 | 61.1 | 5.12E-05 | 20.4 | 8.33E-06 |
| $C_{11}$ | 87.0 | 3.47E-04 | 72.3 | 1.35E-04 | 81.9 | 2.34E-04 | 87.4 | 2.26E-04 | 50.2 | 3.28E-05 | 33.1 | 1.61E-05 |
| $C_{12}$ | 92.9 | 6.73E-04 | 83.4 | 2.60E-04 | 66.4 | 1.02E-04 | 93.0 | 4.30E-04 | 88.5 | 2.51E-04 | 28.1 | 1.28E-05 |
| $C_{13}$ | 95.6 | 1.12E-03 | 82.2 | 2.40E-04 | 65.7 | 9.92E-05 | 96.1 | 8.04E-04 | 87.7 | 2.33E-04 | 46.2 | 2.79E-05 |
| $C_{14}$ | 91.4 | 5.52E-04 | 90.3 | 4.80E-04 | 59.1 | 7.48E-05 | 95.2 | 6.51E-04 | 95.9 | 7.61E-04 | 72.0 | 8.38E-05 |
| $C_{15}$ | 96.7 | 1.53E-03 | 94.5 | 8.98E-04 | 84.4 | 2.80E-04 | 100 | | 96.9 | 1.02E-03 | 83.8 | 1.69E-04 |
| $C_{16}$ | 100 | | 96.7 | 1.41E-03 | 89.0 | 4.18E-04 | 100 | | 96.4 | 8.70E-04 | 88.0 | 2.38E-04 |
| $C_{17}$ | 100 | | 95.1 | 1.00E-03 | 81.5 | 2.28E-04 | 100 | | 96.0 | 7.73E-04 | 88.0 | 2.39E-04 |
| $C_{18}$ | 100 | | 64.6 | 9.44E-05 | 85.0 | 2.93E-04 | 100 | | 92.5 | 4.04E-04 | 100 | |
| $C_{19}$ | 100 | | 100 | | | | 100 | | 100 | | 100 | |
| $C_{20}$ | 100 | | 100 | | | | 100 | | 100 | | | |
| $C_{21}$ | | | 100 | | | | | | 100 | | | |
| $C_{22}$ | | | 100 | | | | | | 100 | | | |
| $C_{23}$ | | | 100 | | | | | | 100 | | | |
| $C_{24}$ | | | | | | | | | 100 | | | |





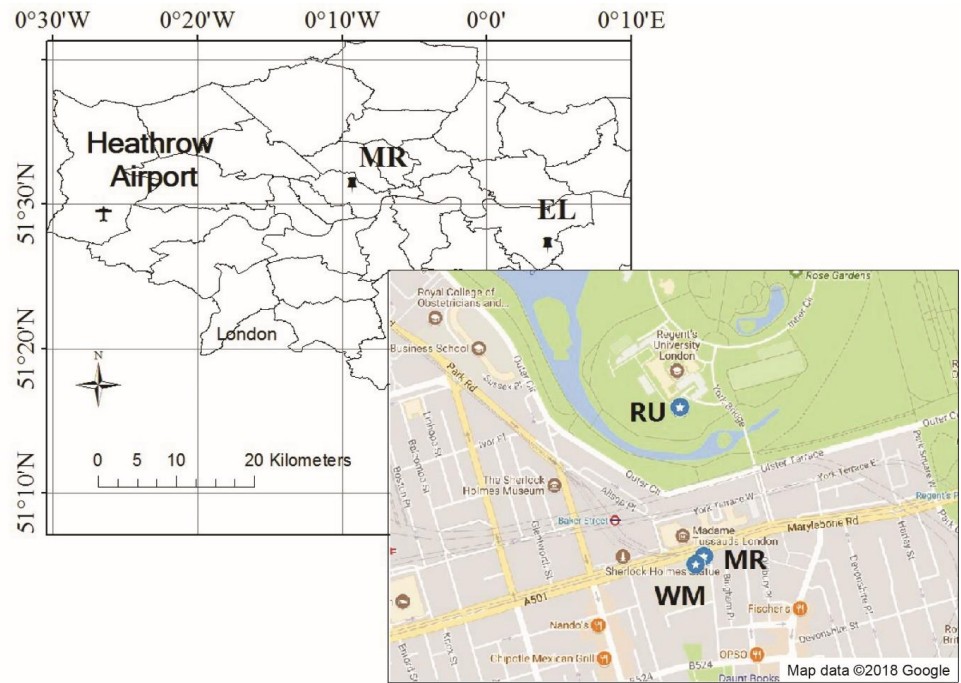

Fig. 1. Map of the sampling sites. RU-Regents University (15 m above ground); WM-University of Westminster (20 m above ground); EL-Eltham; MR-Marylebone Road (south side).




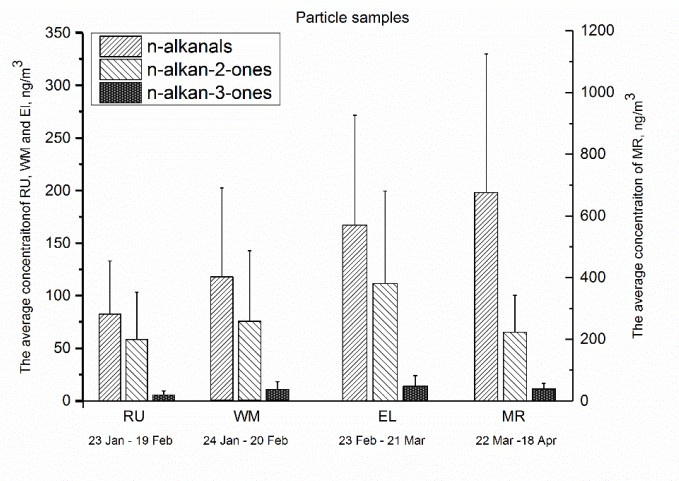

Fig. 2. The average total concentration of particle-bound n-alkanals ($C_8$-$C_{20}$), n-alkan-2-ones ($C_8$-$C_{26}$), and n-alkan-3-ones ($C_8$-$C_{19}$), for each sampling period and site. The error bars indicate one standard deviation.





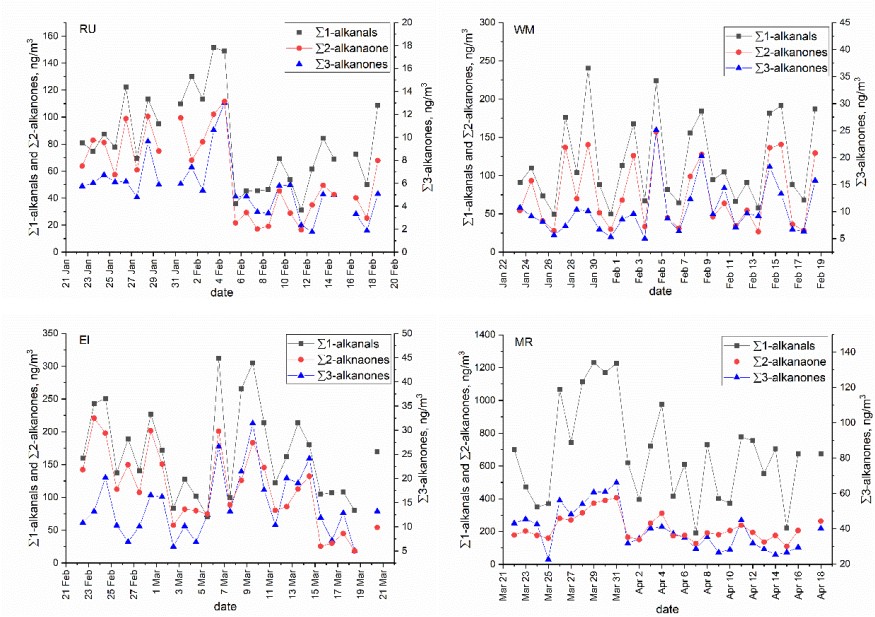

Fig. 3. Time series of particle-bound ∑1-alkanals, ∑n-alkan-2-ones and ∑n-alkan-3-ones at RU, WM, EL, and MR sites.





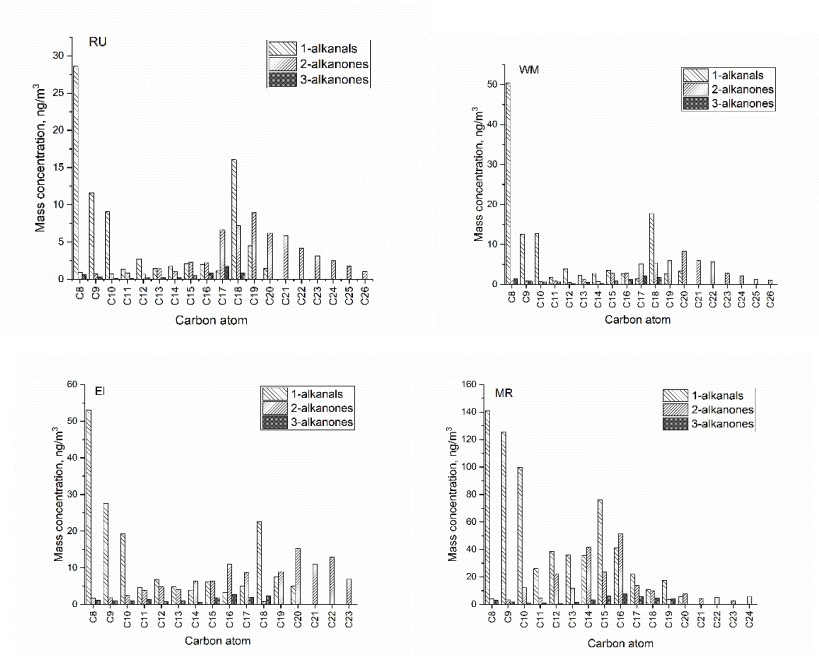

Fig. 4. The molecular distribution of particle-bound carbonyl compounds at four sites (RU, WM, EL, and MR).