# Peer review of "Aliphatic Carbonyl Compounds (C$_8$-C$_{26}$) in Wintertime Atmospheric Aerosol in London, UK"

_Atmospheric Chemistry and Physics, 2018_

## Referee Comment (RC1) · Anonymous Referee #1 · 1 Oct 2018

General Comment:

This manuscript by Lyu et al. describes measurements of three groups of aliphatic carbonyl compounds (n-alkanals, n-alkan-2-ones and n-alkanan-3-ones) in air samples collected in London during winter time. The application of the work sampled at four different sites which included roof-top background, ground-level urban background and street canyon background. The authors found that the concentrations from high to low ordered by n-alkanals, n-alkan-2-ones and n-alkanan-3-ones. Both primary and secondary sources contribute the formation of all compound groups and black carbon and $NO_x$ has relatively low correlation with the products. Vehicle emissions have a strong impact on the air in street canyon location, it is suggested as a major contributor for n-alkanals. Overall, the results are interesting and solid. However, I have some major comments that the authors should address before considered publishable at ACP.

Main Comments:

1. Apparently, the authors have analyzed carbonyl compounds with a limited range of carbon number. The authors should try to provide the range in the abstract or the last paragraph of the introduction. Otherwise, the description at the beginning of the paper is inconsistent with the findings.

2. From Line 207 to Line 247. The manuscript spent a lot of effort comparing results between this study and previous reports. But this part is less well organized and little information if provided in terms of what such a big difference exists.

3. To comprehensively discuss gas-particle partitioning, it is very important to provide information of total organic particle loading at the sampling sites. With that information, one can have a reasonable idea of the fractions of n-alkanes and their products in the particle phase vs. the gas phase. This manuscript starts implying gas-particle partitioning at Line 282, without providing the mass loading information. At typical ambient aerosol loading, C14-C18 n-alkanes should primarily be in the gas phase based on their high vapor pressure. If they observe a > 50% fraction in the particle phase for C14 alkane, it is strongly against the vapor pressure estimates and partitioning theory. It is either from measurement uncertainty or more surprisingly slow evaporation rates after emission from particles. Was this high particle-phase fraction for the "IVOC"-ranged C14-C18 n-alkanes observed at all 4 sites?

4. Starting from Line 286, the manuscript discussed ratios between the n-alkanes, 2-ketones, and 3-ketones, but it is unclear if these ratios are from gas-phase data? Particle-phase data? Or combined? In addition, some conclusions drawn from the ratios, such as the ones at Line 299-302, are not obviously clear. More explanation is needed.

5. CPI usage. The mathematical expression of CPI does not immediately explain what the CPI values mean. The authors should try to provide a little more details, especially information like, what CPI ranges suggesting what sources.

6. Ratios of alkanes/alkanals. The authors compare ratios of C12-C18 alkanes/alkanals at each carbon number between direct diesel vehicle emission data and their particle-phase

data. The similarity between the emission data and the MR site measurement suggests a diesel source of the alkanals at MR. However, it is unclear what the ratios of C8-C10 alkanes/alkanals are compared to and how the authors came to a conclusion of the gasoline source (Line 374-378). In addition, the higher ratios at the other 3 sites may indicate a relatively aged air mass being sampled, as the authors pointed out that alkanals react faster than alkanes. Thus the higher ratios cannot rule out the alkanals at the other sites also have diesel source.

7. Gas-particle partitioning. Line 451-452. It is problematic to assume this. Based on the SIMPOL.1 estiamtes of vapor pressure, C16 alkanal has a C* of 75 ug/m$^3$, and C19 alkanone has a C* of 11 ug/m$^3$. These species are in the SVOC range and should have substantial fraction in the gas phase.

Minor Comments:

Line 70. Should be "…an important source of aliphatic carbonyl"

Line 117. Change "adsorption tubes" to "sorbent tubes" to be consistent with the context.

Line 188 and 194. The same information was repeated twice.

Line 225-226. A reference is needed here.

Line 330. Cmax is defined after already used a few times. The same for CPI.

Line 249-253. These discussion should be moved before Line 229.

Line 270. It is unclear from these two references that whether OH quickly attacks H at the one position.

Section 3.2 is too short to be an individual topic. Not much discussion is on this part anyway. Suggest merge it into other sections.

Line 413-414. How can a "moderate" correlation indicate a "substantial" source?

---

## Referee Comment (RC2) · Anonymous Referee #2 · 11 Oct 2018

Review of, "Aliphatic Carbonyl Compounds (C8-C26) in Wintertime Atmopsheric Aerosol in London, UK"

General Comments:

This study provides measurements of three groups of carbonyls: n-alkanals, n-alkan-2-ones, and n-alkan-3-ones across a wide range of carbon numbers in both the gas and particle phases at one urban and three background sites of London. The n-alkanal concentrations were observed to be the highest at all sites, followed by those of the n-alkaln-2-ones, and n-alkan-3-ones. Homologue distributions are presented and tracer correlations are explored to infer anthropogenic emissions as the primary source for alkanals. Empirical gas-particle partitioning coefficients are also provided. While generally this dataset has value and would be of interest to ACP readership, the manuscript's writing needs to be greatly improved before publication. Improvements in terms of organization, focus, and precision of discussions when comparing to previous literature are suggested in the specific comments below. Regarding organization, authors should consider reordering some sections as results or statements are made as fact without support until much later in the manuscript (e.g. alkanals are said early on to be from anthropogenic emissions, yet measurements and analysis support of this are discussed near end).

Specific Comments:

1. Lines 117-118: What were the recovery efficiencies? Was breakthrough of the PTFE filters addressed? Specifically, semi-volatile components in particles that make it to the sorbent tubes?
2. Line 246: CPI has not been introduced properly to discuss here out of context
3. Presentation of literature should be more precisely worded regarding use of Zhang, Ruehl, Schilling Fahnestock, and Yee et al. references:
   a. Line 74-75: Add Yee et al., 2012 with this group.
   b. Only reference Zhang et al., 2015 and Ruehl et al., 2013 positively identify carbon position of the carbonyl groups. Other references sum isomers together/propose structures of compounds with some of the ketone group positions listed in lines 80-82, but they were not specifically isolated as authors suggest. Probably better to simply delete those lines.
      i. Lines 80-82 should be revised to read more along the lines of, "...chamber studies of dodecane oxidation include observation of aldehydes and ketones as oxidation products...".
      ii. In lines 250-253, to generally say that these compounds with "few carbon atoms are believed mainly to originate as the fragmental products from n-alkanes" and that "higher compounds are mainly generated from functional pathways" as an extension to the atmosphere is not actually supported by these references. Further, what is the cutoff for "few carbon atoms"? It seems that the authors instead are inferring this in the context of their results. It may be possible for their measurements to address this question in fact, which would be interesting and should be brought to more focus in the Introduction if so. Authors should at minimum revise the wording to "Carbonyls including n-alkan-2-one and n-alkan-3one homologues could result as fragmentation products from larger alkane precursors during gas-phase oxidation (Yee et al., 2012; Schilling-

Fahnestock et al., 2015) or as functionalized products from heterogeneous oxidation of particle-bound alkanes (Ruehl et al., 2013; Zhang et al., 2015).

4. Lines 260-278: This discussion seems more relevant to put in the introduction as motivation for why measurement of carbonyls and the specific carbon position of ketones is important. If the authors can restructure the writing, it seems that they are trying to utilize their measurements to infer sources of the measured carbonyls from homogenous/gas-phase oxidation and heterogeneous oxidation which is told by ketone number position. Though, this is not rigorously addressed using the measurements in the same way Zhang et al., 2015 do. So, either limit the specificity on the literature that is presented here and change language throughout the manuscript to "lightly" suggest chamber and flow tube measurements as supporting the trends in the presented measurements or do a more rigorous analysis to focus on the phase of oxidation and ratios of ketone carbon number position. This is further difficult to address with the measurements presented as is because there are no n-alkane distributions provided. The authors need to adjust the certainty in language used when describing that something must derive from gas or heterogeneous oxidation.

5. Line 280: Are there references that can be added to support anthropogenic activities as a source of aldehydes and to what degree? Cite Table S3 here. This becomes addressed later in the manuscript, so it seems odd to state with such certainty early-on without providing the measurements and discussions up front.

6. Line 284: Where do these numbers come from? Include the particulate form % for the low MW n-alkanes here to compare with that of C14-C36. Why are these not included along in an SI Table like Table 2 or with Table 2?

7. Lines 265, 288, 297 (and any others throughout manuscript): Replace "homogeneous" with "gas-phase". Homogeneous reaction should not be used to synonymously refer to the gas-phase reaction of alkane (gas) + OH (gas). One could have a homogeneous reaction in the particle phase as well (both reactants are in the particle phase). Heterogeneous reaction across phases: OH (gas) reacting with alkane (particle phase) in some contexts presented in the manuscript.

8. Lines 287-289: The authors should provide context as to what these ratios mean, what ranges are expected for meaning what (primarily gas-phase vs heterogeneous oxidation, etc.).

9. Lines 289-290: Provide additional support from literature or from the conducted measurements for this claim of 2-ketones being from primary emission sources that is supported. For example, Table S3 provides some support for alkanals correlation with BC and NOx, but why is this analysis not done with the 2-ketones as well? Correlations are provided much later in manuscript. Why not provide similar Table S3 for ketone groups?

10. Lines 290-292: Are the photochemical conditions (e.g. NOx conditions) during the field studies close enough to those of the cited chamber studies (low-NOx) from Yee et al. 2012 and Schilling Fahnestock et al., 2015 to be applicable here as plausible mechanisms? It seems that in non-rural environments, well-established mechanisms of ketone formation as in Lim et al., 2009 for alkane oxidation in the presence of NOx would be another/more applicable route of formation to explain these products. Further, what are the actual NOx levels in the current study?

Reference:

Lim, Yong Bin and Ziemann, Paul J. 'Chemistry of Secondary Organic Aerosol Formation from OH Radical-Initiated Reactions of Linear, Branched, and Cyclic Alkanes in the Presence of NO$x$', Aerosol Science and Technology, 43: 6, 604-619, 2009.

11. Line 303: This claim is too strong as the measurements might be indicative of such, but there are no measurements that actually verify this. Change "…were attributed to…", to "might be explained by" and "…reactions were expected…" to "would be the expected dominant pathway."
12. Section 3.2: This is oddly placed and should really be placed at the beginning of the Results and Discussion section as Section 3.1. Figure 3 should come before Figure 2.
13. Line 330: It would be beneficial to the readers to include a brief sentence orienting the range in CPI values that is traditionally assumed to be indicative of anthropogenic/biogenic sources. Same with Cmax. Also, lines 330-331 do not make sense as written. Cmax is merely the carbon number of the carbonyl with the highest concentration, correct?
14. Lines 337-338: Seems that in arguing for any carbonyl to be an oxidation product, fragment, or primary emission, the authors should present Cmax and CPI values for distributions of regular n-alkanes.
15. Lines 380-384: This paragraph is out of place and does not provide value to the manuscript (at least is not further placed in context or expanded upon).
16. Lines 388-390: Not sure what is meant here.
17. Lines 427-438: Do the diesel engine laboratory tests show indications of ketones in the exhaust as well to support some of the hypotheses made here?
18. Lines 451-453: Why was this assumption made when measurements of these compounds were actually performed to get empirical Kp?
19. Table 2: It would be helpful in interpretation of the results here to show TSP values as well (For example: why is % in particle phase higher for similar carbon #'s at EL site?)
20. Lines 487-491: These sentences seem contradictory pointing to heterogeneous n-alkane oxidation vs anthropogenic primary sources as origin of the ketones. Perhaps just language needs to be changed to not make it seem like one source dominates over the other.
21. The conclusions section is written more like a results and discussion section with specific correlations, ratios, and comparisons of these ratios to fuel sources. Rewrite to focus more on bigger implications. For example, what are the implications of 2-ketones regression of log Kp vs vapor pressure having a better fit compared to the alkanals and the 3-ketones? Seems like this should say something about equilibrium vs non-equilibrium conditions and the timescales of aging, oxidation, and partitioning. Is this indicative of specific chemical pathways and/or uncertainties not properly accounted for the alkanals and 3-ketones source?

Technical Corrections:

1. No need to repeat lines 176-178 from lines 141-143.
2. Lines 192-194: Seems unnecessary to make this separate paragraph.
3. It does not seem convention to specify the C1 position for aldehyde names as in:
   a. Line 80, "1-undecanal" should just be undecanal
   b. Figure 3 caption. "1-" for alkanals
   c. Line 280

4. Line 274: Change subscripted reference 18 to proper format.
5. Line 355: repetitive Cmax information on MR can be deleted.
6. Line 440: Suggest renaming this section to "Gas and Particle Phase Partitioning"

---

## Author Comment (AC1) · 18 Dec 2018

Journal: ACP Title: Aliphatic Carbonyl Compounds (C8–C26) in Wintertime Atmospheric Aerosol in London, UK Author(s): Ruihe Lyu et al. MS No.: acp-2018-769

RESPONSE TO REVIEWERS

We thank the reviewers for their very detailed and insightful reviews which have led to many enhancements to the paper. A marked up copy of the changes made are in the attached Supplement.

REVIEWER #1 General Comment: This manuscript by Lyu et al. describes measurements of three groups of aliphatic carbonyl compounds (n-alkanals, n-alkan-2-ones and n-alkanan-3-ones) in air samples collected in London during winter time. The application of the work sampled at four different sites which included roof-top background, ground-level urban background and street canyon background. The authors found that the concentrations from high to low ordered by n-alkanals, n-alkan-2-ones and n-alkanan-3-ones. Both primary and secondary sources contribute the formation of all compound groups and black carbon and NOx has relatively low correlation with the products. Vehicle emissions have a strong impact on the air in street canyon location, it is suggested as a major contributor for n-alkanals. Overall, the results are interesting and solid. However, I have some major comments that the authors should address before considered publishable at ACP. Main Comments: 1. Apparently, the authors have analyzed carbonyl compounds with a limited range of carbon number. The authors should try to provide the range in the abstract or the last paragraph of the introduction. Otherwise, the description at the beginning of the paper is inconsistent with the findings. RESPONSE: The carbon number range of carbonyl compounds is now described in the abstract.

2. From Line 207 to Line 247. The manuscript spent a lot of effort comparing results between this study and previous reports. But this part is less well organized and little information if provided in terms of what such a big difference exists. RESPONSE: The text has been lightly edited, and explanations for the differences from earlier work provided.

3. To comprehensively discuss gas-particle partitioning, it is very important to provide information of total organic particle loading at the sampling sites. With that information, one can have a reasonable idea of the fractions of n-alkanes and their products in the particle phase vs. the gas phase. This manuscript starts implying gas-particle partitioning at Line 282, without providing the mass loading information. At typical ambient aerosol loading, C14-C18 n-alkanes should primarily be in the gas phase based on their high vapor pressure. If they observe a > 50% fraction in the particle phase for C14 alkane, it is strongly against the vapor pressure estimates and partitioning theory. It is either from measurement uncertainty or more surprisingly slow evaporation rates after emission from particles. Was this high particle-phase fraction for the "IVOC"-ranged C14-C18 n-alkanes observed at all 4 sites? RESPONSE: We regret that an incorrect dataset for the condensed phase was described in the first version of the paper. This has now been replaced with new data, and PM mass loading data are also provided in Table S5. 4. Starting from Line 286, the manuscript discussed ratios between the n-alkanes, 2-ňketones, and 3-ketones, but it is unclear if these ratios are from gas-phase data? Particle-phase data? Or combined? In addition, some conclusions drawn from the ratios, such as the ones at Line 299-302, are not obviously clear. More explanation is needed. RESPONSE: The ratios between n-alkanes, 2-Âňketones, and 3-ketones were from the sum of concentrations of gas and particle phases. This is now clarified in the text.

Line 299-302, RESPONSE: This sentence has been re-worded to provide clarification. 5. CPI usage. The mathematical expression of CPI does not immediately explain what the CPI values mean. The authors should try to provide a little more details, especially information like, what CPI ranges suggesting what sources. RESPONSE: An explanatory sentence has been added.

6. Ratios of alkanes/alkanals. The authors compare ratios of C12-C18 alkanes/alkanals at each carbon number between direct diesel vehicle emission data and their particle-phase data. The similarity between the emission data and the MR site measurement suggests a diesel source of the alkanals at MR. However, it is unclear what the ratios of C8-C10 alkanes/alkanals are compared to and how the authors came to a conclusion of the gasoline source (Line 374-378). In addition, the higher ratios at the other 3 sites may indicate a relatively aged air mass being sampled, as the authors pointed out that alkanals react faster than alkanes. Thus the higher ratios cannot rule out the alkanals at the other sites also have diesel source. RESPONSE: A reference to literature data for C8-C10 compounds from gasoline engines is now included. Commentary on the higher ratios at other sites is now added.

7. Gas-particle partitioning. Line 451-452. It is problematic to assume this. Based on the SIMPOL.1 estimates of vapor pressure, C16 alkanal has a C* of 75 ug/m3, and C19 alkanone has a C* of 11 ug/m3. These species are in the SVOC range and should have substantial fraction in the gas phase. RESPONSE: We thank the reviewer for raising this. The vapor concentrations are unlikely to be zero, but were below detection limit. An explanatory note has been added at the bottom of Table 2.

Minor Comments: Line 70. Should be "...an important source of aliphatic carbonyl" RESPONSE: Amended as suggested. Line 117. Change "adsorption tubes" to "sorbent tubes" to be consistent with the context. RESPONSE: Amended as suggested.

Line 188 and 194. The same information was repeated twice. RESPONSE: The second expression of this information has been deleted.

Line 225-226. A reference is needed here. RESPONSE: The following references are now cited: Simoneit, B. R. T., Cox, R. E., and Standley, L. J.: Organic matter of the troposphere - IV. Lipids in Harmattan aerosols of Nigeria, Atmos. Environ., 22, 983-1004, https://doi.org/10.1016/0004-6981(88)90276-4, 1967. Gogou, A., Stratigakis, N., Kanakidou, M., and Stephanou, E. G.: Organic aerosols in Eastern Mediterranean: components source reconciliation by using molecular markers and atmospheric back trajectories, Org. Geochem., 25, 79-96, https://doi.org/10.1016/S0146-6380(96)00105-2,1996.

Line 330. Cmax is defined after already used a few times. The same for CPI. RESPONSE: We do not feel that this is problematic, and may occasionally assist the reader (not all of which start reading at page 1).

Line 249-253. These discussion should be moved before Line 229. RESPONSE: Moved as recommended. Line 270. It is unclear from these two references that whether OH quickly attacks H at the one position. RESPONSE: This has been clarified by removing the word "quickly".

Section 3.2 is too short to be an individual topic. Not much discussion is on this part anyway. Suggest merge it into other sections. RESPONSE: Agreed. Now merged into 3.1.

Line 413-414. How can a "moderate" correlation indicate a "substantial" source? RESPONSE: The sentence has been modified to resolve this contradiction.

REVIEWER #2 Review of, "Aliphatic Carbonyl Compounds (C8‐C26) in Wintertime Atmopsheric Aerosol in London, UK" General Comments: This study provides measurements of three groups of carbonyls: n‐alkanals, n‐alkan‐2‐ones, and n‐alkan‐3‐ones across a wide range of carbon numbers in both the gas and particle phases at one urban and three background sites of London. The n‐alkanal concentrations were observed to be the highest at all sites, followed by those of the n‐alkaln‐2‐ones, and n‐alkan‐3‐ones. Homologue distributions are presented and tracer correlations are explored to infer anthropogenic emissions as the primary source for alkanals. Empirical gas‐particle partitioning coefficients are also provided. While generally this dataset has value and would be of interest to ACP readership, the manuscript's writing needs to be greatly improved before publication. Improvements in terms of organization, focus, and precision of discussions when comparing to previous literature are suggested in the specific comments below. Regarding organization, authors should consider reordering some sections as results or statements are made as fact without support until much later in the manuscript (e.g. alkanals are said early on to be from anthropogenic emissions, yet measurements and analysis support of this are discussed near end).

Specific Comments: 1. Lines 117‐118: What were the recovery efficiencies? Was breakthrough of the PTFE filters addressed? Specifically, semi‐volatile components in particles that make it to the sorbent tubes? RESPONSE: Filter artefacts were not evaluated, but this kind of sampling train is in widespread use. Provided pressure drops are modest, loss from the particles should be minimal. A breakthrough test was made on the sorption tubes and showed a minimal breakthrough for alkanes $\geq$C11.

2. Line 246: CPI has not been introduced properly to discuss here out of context. RESPONSE: An explanation of CPI values is now included.

3. Presentation of literature should be more precisely worded regarding use of Zhang, Ruehl, Schilling Fahnestock, and Yee et al. references: a. Line 74‐75: Add Yee et al., 2012 with this group. RESPONSE: This reference has been added.

b. Only reference Zhang et al., 2015 and Ruehl et al., 2013 positively identify carbon position of the carbonyl groups. Other references sum isomers together/propose structures of compounds with some of the ketone group positions listed in lines 80‐82, but they were not specifically isolated as authors suggest. Probably better to simply delete those lines. RESPONSE: The lines have been amended to indicate that carbon positions of carbonyl groups were not identified in all studies.

i. Lines 80‐82 should be revised to read more along the lines of, ". . .chamber studies of dodecane oxidation include observation of aldehydes and ketones as oxidation products. . .". RESPONSE: Amended as recommended.

ii. In lines 250‐253, to generally say that these compounds with "few carbon atoms are believed mainly to originate as the fragmental products from n‐alkanes" and that "higher compounds are mainly generated from functional pathways" as an extension to the atmosphere is not actually supported by these references. Further, what is the cutoff for "few carbon atoms"? It seems that the authors instead are inferring this in the context of their results. It may be possible for their measurements to address this question in fact, which would be interesting and should be brought to more focus in the Introduction if so. Authors should at minimum revise the wording to "Carbonyls including n‐alkan‐2‐one and nÂňalkan‐3one homologues could result as fragmentation products from larger alkane precursors during gas‐phase oxidation (Yee et al., 2012; SchillingÂňFahnestock et al., 2015) or as functionalized products from heterogeneous oxidation of particle‐bound alkanes (Ruehl et al., 2013; Zhang et al., 2015)." RESPONSE: The suggested wording has been added to the text.

[Figure]

4. Lines 260‐278: This discussion seems more relevant to put in the introduction as motivation for why measurement of carbonyls and the specific carbon position of ketones is important. If the authors can restructure the writing, it seems that they are trying to utilize their measurements to infer sources of the measured carbonyls from homogenous/gas‐phase oxidation and heterogeneous oxidation which is told by ketone number position. Though, this is not rigorously addressed using the measurements in the same way Zhang et al., 2015 do. So, either limit the specificity on the literature that is presented here and change language throughout the manuscript to "lightly" suggest chamber and flow tube measurements as supporting the trends in the presented measurements or do a more rigorous analysis to focus on the phase of oxidation and ratios of ketone carbon number position. This is further difficult to address with the measurements presented as is because there are no n‐alkane distributions provided. The authors need to adjust the certainty in language used when describing that something must derive from gas or heterogeneous oxidation. RESPONSE: The text has been substantially revised.

5. Line 280: Are there references that can be added to support anthropogenic activities as a source of aldehydes and to what degree? Cite Table S3 here. This becomes addressed later in the manuscript, so it seems odd to state with such certainty early‐on without providing the measurements and discussions up front. RESPONSE: Lines 280-282 have been revised to include references.

6. Line 284: Where do these numbers come from? Include the particulate form % for the low MW n‐alkanes here to compare with that of C14‐C36. Why are these not included along in an SI Table like Table 2 or with Table 2? RESPONSE: The n-alkane data will be published elsewhere. The text has been amended to reflect this.

7. Lines 265, 288, 297 (and any others throughout manuscript): Replace "homogeneous" with "gas‐phase". Homogeneous reaction should not be used to synonymously refer to the gas‐phase reaction of alkane (gas) + OH (gas). One could have a homogeneous reaction in the particle phase as well (both reactants are in the particle phase). Heterogeneous reaction across phases: OH (gas) reacting with alkane (particle phase) in some contexts presented in the manuscript. RESPONSE: Amended as recommended.

8. Lines 287‐289: The authors should provide context as to what these ratios mean, what ranges are expected for meaning what (primarily gas‐phase vs heterogeneous oxidation, etc.). RESPONSE: As implied by the previous sentence, a ratio of >1 can be taken to imply a heterogeneous mechanism in the absence of primary sources. 9. Lines 289‐290: Provide additional support from literature or from the conducted measurements for this claim of 2‐ketones being from primary emission sources that are supported. For example, Table S3 provides some support for alkanals correlation with BC and NOx, but why is this analysis not done with the 2‐ketones as well? Correlations are provided much later in manuscript. Why not provide similar Table S3 for ketone groups? RESPONSE: A reference to primary emissions has been added at this point. The section on correlations addresses 3-ketones as well as 2-ketones. See Section 3.2.3. 10. Lines 290‐292: Are the photochemical conditions (e.g. NOx conditions) during the field studies close enough to those of the cited chamber studies (low‐NOx) from Yee et al. 2012 and Schilling Fahnestock et al., 2015 to be applicable here as plausible mechanisms? It seems that in non‐rural environments, well‐established mechanisms of ketone formation as in Lim et al., 2009 for alkane oxidation in the presence of NOx would be another/more applicable route of formation to explain these products. Further, what are the actual NOx levels in the current study? RESPONSE: The average NOx concentrations were EL(23.35 $\mu$g/m3), MR(202 $\mu$g/m3) and were not measured at the RU and WM sites. The concentrations of NOx (RU, WM, EL, MR) during our sampling period were between the low-NOx (Yee et al., 2012; Schilling Fahnestock et al., 2015) and high-NOx condition (Lim and Ziemann, 2009). Additional text is now included which discusses the mechanistic implications of the presence of NO, and now includes reference to the work of Lim and Ziemann.

11. Line 303: This claim is too strong as the measurements might be indicative of such, but there are no measurements that actually verify this. Change "...were attributed to...", to "might be explained by" and "...reactions were expected..." to "would be the expected dominant pathway." RESPONSE: Agreed. Amended as suggested.

12. Section 3.2: This is oddly placed and should really be placed at the beginning of the Results and Discussion section as Section 3.1. Figure 3 should come before Figure 2. RESPONSE: Moved, as recommended.

13. Line 330: It would be beneficial to the readers to include a brief sentence orienting the range in CPI values that is traditionally assumed to be indicative of anthropogenic/biogenic sources. Same with Cmax. Also, lines 330‐331 do not make sense as written. Cmax is merely the carbon number of the carbonyl with the highest concentration, correct? RESPONSE: A brief explanation of CPI has been added. We thank the reviewer for pointing out the meaningless nature of lines 330-331 and have now amended the sentence.

14. Lines 337‐338: Seems that in arguing for any carbonyl to be an oxidation product, fragment, or primary emission, the authors should present Cmax and CPI values for distributions of regular n‐alkanes. RESPONSE: The n-alkane data will be presented elsewhere. A sentence has been added to summarise the homologue distributions (Cmax) and CPI values) of the n-alkanes.

15. Lines 380‐384: This paragraph is out of place and does not provide value to the manuscript (at least is not further placed in context or expanded upon). RESPONSE: This material has been edited and moved to the first paragraph of this section.

16. Lines 388‐390: Not sure what is meant here. RESPONSE: Now amended to clarify. 17. Lines 427‐438: Do the diesel engine laboratory tests show indications of ketones in the exhaust as well to support some of the hypotheses made here? RESPONSE: Ketones were below detection limit in the diesel exhaust. This information has been added to this section.

18. Lines 451‐453: Why was this assumption made when measurements of these compounds were actually performed to get empirical Kp? RESPONSE: This statement has been modified in the light of comments from both reviewers.

19. Table 2: It would be helpful in interpretation of the results here to show TSP values as well (For example: why is % in particle phase higher for similar carbon #'s at EL site?) RESPONSE: The values are as below. These have been added to the SI. We have an explanation for the % particle phase being higher at EL.

Sites PM10 Range, $\mu$g/m3 PM10 Mean $\mu$g/m3 Note RU and WM 10.8-72.4 34.1 The sampling period was dominated by southerly winds and the data from London, North Kensington were used as this is an upwind urban background site. EL 4.37-27.1 19.3 The PM10 data was obtained from the London North Kensington site (Defra), because the EL only have PM2.5 data, and the PM2.5 data of two site (EL and London North Kensington) were close to each other. MR 12.6-78.7 30.7 MR site

20. Lines 487‐491: These sentences seem contradictory pointing to heterogeneous n‐alkane oxidation vs anthropogenic primary sources as origin of the ketones. Perhaps just language needs to be changed to not make it seem like one source dominates over the other. RESPONSE: We agree that this was contradictory and have amended it to provide greater clarity.

21. The conclusions section is written more like a results and discussion section with specific correlations, ratios, and comparisons of these ratios to fuel sources. Rewrite to focus more on bigger implications. For example, what are the implications of 2‐ketones regression of log Kp vs vapor pressure having a better fit compared to the alkanals and the 3‐ketones? Seems like this should say something about equilibrium vs non‐equilibrium conditions and the timescales of aging, oxidation, and partitioning. Is this indicative of specific chemical pathways and/or uncertainties not properly accounted for the alkanals and 3‐ketones source? RESPONSE: The reviewer raised a good point for which we add new text.

Technical Corrections: 1. No need to repeat lines 176‐178 from lines 141‐143. RESPONSE: Agreed and amended.

2. Lines 192‐194: Seems unnecessary to make this separate paragraph. RESPONSE: Agreed and amended.

3. It does not seem convention to specify the C1 position for aldehyde names as in: a. Line 80, "1‐undecanal" should just be undecanal b. Figure 3 caption. "1‐" for alkanals c. Line 280 RESPONSE: Agreed and amended. 4. Line 274: Change subscripted reference 18 to proper format. RESPONSE: Agreed and amended.

5. Line 355: repetitive Cmax information on MR can be deleted. RESPONSE: We do not find any repetition.

6. Line 440: Suggest renaming this section to "Gas and Particle Phase Partitioning" RESPONSE: Agreed and amended.

Please also note the supplement to this comment:
https://www.atmos-chem-phys-discuss.net/acp-2018-769/acp-2018-769-AC1-supplement.pdf

**Supplement:**

[revised manuscript text omitted]

---

## Author Response (AR2)

**Journal: ACP**
**Title: Aliphatic Carbonyl Compounds (C8–C26) in Wintertime Atmospheric Aerosol in London, UK**
**Author(s): Ruihe Lyu et al.**
**MS No.: acp-2018-769**

**RESPONSE TO CO-EDITOR**

Sect 3.3: The log(Kp) vs log(VPt) results should be discussed in more detail. For instance, according to Pankow (1994) the slope of this plot should be about -1. For the alkanals, the slope is sometimes positive. Please discuss. Since samples were collected for 24 h, how was the temperature determined? How does the diurnal evolution of temperature affect the results?
**RESPONSE:**   New text has been added as follows:

According to theory, the gradient of the plot of log $K_P$ versus log ($VP_T$) should be -1 (Pankow, 1994). However, many measurement datasets for a number of semi-volatile compound groups including n-alkanes (Cincinelli et al., 2007; Karanasiou et al., 2007; Mandalakis et al., 2002) and PAH (Callen et al., 2008; Wang et al., 2011; Ma et al., 2011; Mandalakis et al., 2002) show a range of values of gradient, often around -0.5, but ranging to below -1, and in some cases positive.   Callen et al., (2008) discuss the reasons for deviation from a value of -1, which include a lack of equilibrium, absorption into the organic matter (shallower than -0.6), adsorption processes (steeper than -1), and the averaging of conditions across a range of temperatures during a sampling period.

Our data for alkan-2-ones show high $r^2$ values and values of gradient (m) in the range of the literature for other groups of semi-volatile compounds.   Average gradients at the four sites ranged from -0.46 to -0.26.   The alkan-3-ones show generally considerably lower values of $r^2$ and average values of gradient at the four sites of -0.43 to -0.23.   This poorer correlation could be the result of lower analytical precision.   The n-alkanals show still lower values of $r^2$, and more variable and shallower values of slope. Mean slopes for the four sites ranged from -0.23 to -0.16.   There were no positive daily values.   The lower $r^2$ may be a result of disequilibrium for the alkanals which are dominated by primary emissions, and are also more reactive. It might also reflect a role for aqueous aerosol as an absorbing medium for these compounds containing a significant polar moiety, which would lead to deviations from the Pankow (1994) theory, and more variable behaviour as the availability of aqueous particles into which to partition would depend upon relative humidity, which is itself highly variable.

Samples were collected over 24-hour periods and hence the diurnal variation of temperature may be relevant.   Temperature data were taken from Heathrow Airport to the west of London and did not show large diurnal fluctuations, so this should not be a major factor.   The average diurnal temperature range based upon hourly data was 6.9⁰C.

Table 2: The % of n-alkanals in the particle phase is surprisingly high for the low C number species. Please comment on this.
**RESPONSE:**   We have re-evaluated our recovery data and have excluded vapour phase compounds of $C_8$ and $C_9$ from the data analysis as these were not collected quantitatively.

Minor:
Abstract: Please make it clear that both gas- and condensed-phase measurements were made as part of this work.
**RESPONSE:**   This has been accomplished.

Sect 2.3: Please also include limit of detections for the compounds as this becomes a point later on in the manuscript. Also, please add information on breakthrough that was included in the response to Referee 2. What exactly is meant by "minimal breakthrough for alkanes >= C11"? Was loss of gas-phase compounds to the filters investigated?

RESPONSE:   Limits of detection have been added to the Supplementary Information.   Six tests of vapour breakthrough were conducted with two adsorption tubes in series.   Recovery was good for all compounds $\geq C_{11}$, and for $C_{10}$ was 85% for all three compound groups. This is now clarified in the text. No tests of vapour uptake to the filters were conducted, but as PTFE filters were used uptake is not to be expected (unlike quartz).

Line 208: The time series is Figure 3 not 2. Please ensure that all figures are referenced in order and that the correct figure is referenced in the text throughout the manuscript.
**RESPONSE:**   This has been corrected.

Line 214-216: These references did not specifically identify the position of the carbonyl group. Please revise.
**RESPONSE:**   Revised to clarify the point.

Line 228-229: Wouldn't this be independent of NO concentration if the issue is reaction with O2 vs isomerization?
**RESPONSE:**   There was a typographic error which has been corrected.   The role of NO is in converting the alkylperoxy radical to an alkoxy radical which isomerises, rather than reacting with $O_2$.

Line 252-253: I am not convinced that the difference in concentration is a "clear indication of road traffic." I would expect that the urban measurements would also be influenced by traffic. More details regarding the previous work (site locations, temperature, time of year, etc.) needs to be provided if this sentence remains.
**RESPONSE:**   An explanatory sentence has been added as follows:

Earlier work has clearly demonstrated a substantial elevation in traffic-generated pollutants at the Marylebone Road site, relative to background sites within London (Harrison and Beddows, 2017).

Line 261: Please define Cmax the first time it is used.
**RESPONSE:**   This has been amended.

Line 286: CPI is used before it is defined. Please correct this.
**RESPONSE:**   This has been amended.

Line 347-348: Please be more explicit about what is meant by "significant particulate fraction" and "low MW."
**RESPONSE:**   This refers to a particulate fraction >60% and n-alkanes of $C_{14}$-$C_{18}$, and the text has been amended.

Line 355-357: For alkanes of this size, isomerization in the gas-phase dominates over fragmentation. As such I would think that decomposition products would likely be multifunctional. I would also expect that aldehydes would be preferentially formed as a result of fragmentation. Please clarify how this could occur.
**RESPONSE:**   The word "likely" has been changed to "possible", and the following has been added to the sentence:

"….although fragmentation reactions would result mainly in the formation of alkanals, and are less likely to occur than isomerisation leading mostly to multifunctional products.".

Line 365-366: Please clarify that homogeneous means gas-phase.
**RESPONSE:**   This has been clarified.

Table S4 I see data for only RU and WM sites, however, the caption says that it includes data for El and MR site. Please fix.

**RESPONSE:** The full dataset for all sites is now included.

Lines 582-588: This seems better suited for Sect. 3.3 then for the conclusions.
**RESPONSE:** This is within the new text in Section 3.3, and has therefore been shortened to a summary within the Conclusions.

[revised manuscript text omitted]

**RU** (Regent University, 15 m above ground, 23 Jan - 19 Feb 2017)

[revised manuscript text omitted]

Table S2S3. The ratios of n-alkan-2-ones/n-alkanes, and n-alkan-3-ones/n-alkanes with same
carbon numbers.

| n-alkan-2-ones/n-alkanes, % | | | | |
|---|---|---|---|---|
| Carbon numbers | RU | WM | EL | MR |
| | | | | Northerly winds | Southerly winds |
| C13 | 2.31 | 1.86 | 2.14 | 6.02 | 6.34 |
| C14 | 2.03 | 1.58 | 2.38 | 20.8 | 15.9 |
| C15 | 4.17 | 4.77 | 5.95 | 11.3 | 13.6 |
| C16 | 7.35 | 5.98 | 8.58 | 11.8 | 7.27 |
| C17 | 48.9 | 29.8 | 31.7 | 27.0 | 24.6 |
| C18 | 32.5 | 18.2 | 31.3 | 23.4 | 18.1 |
| C19 | 77.8 | 33.9 | 43.2 | 19.6 | 14.7 |
| C20 | 379 | 168 | 33.3 | 32.8 | 10.1 |
| C21 | 267 | 59.3 | 33.1 | 10.4 | 3.18 |
| C22 | 144 | 36.1 | 18.6 | 4.76 | 5.29 |
| C23 | 17.1 | 6.61 | 10.5 | 1.75 | 3.41 |
| C24 | 17.1 | 7.16 | 11.3 | 3.31 | 5.62 |
| C25 | 26.5 | 4.24 | | | |
| C26 | 18.3 | 3.36 | | | |
| averages | 74.7 | 27.2 | 19.3 | 14.4 | 10.1 |
| min | 2.03 | 1.58 | 2.14 | 1.75 | 3.18 |
| max | 379 | 168 | 43.2 | 32.8 | 24.6 |

| n-alkan-3-ones/n-alkanes, % | | | | |
|---|---|---|---|---|
| Carbon numbers | RU | WM | EL | MR |
| | | | | Northerly winds | Southerly winds |
| C13 | 0.70 | 0.50 | 0.49 | 1.30 | 1.39 |
| C14 | 0.58 | 0.50 | 0.47 | 2.10 | 1.73 |
| C15 | 1.31 | 1.51 | 2.15 | 5.22 | 3.58 |
| C16 | 3.80 | 1.73 | 3.87 | 14.9 | 4.44 |
| C17 | 12.4 | 7.17 | 8.39 | 15.7 | 8.58 |
| C18 | 3.78 | 3.87 | 7.61 | 11.1 | 3.59 |
| C19 | | | | 7.65 | 8.27 |
| averages | 3.76 | 2.55 | 3.83 | 8.28 | 4.51 |
| min | 0.58 | 0.50 | 0.47 | 1.30 | 1.39 |
| max | 12.4 | 7.17 | 8.39 | 15.7 | 8.58 |

Table S3S4. The regression equations between black carbon (BC) and n-alkanals, NO$_x$ and n-alkanals.

$C_{\text{n-alkanals}} = m\, C_{BC} + b$

| Carbon atoms | RU | | | WM | | | MR | | | | | |
|---|---|---|---|---|---|---|---|---|---|---|---|---|
| | | | | | | | Northerly Winds | | | Southerly winds | | |
| | m | b | r$^2$ | m | b | r$^2$ | m | b | r$^2$ | m | b | r$^2$ |
| C8 | -0.43 | 34.3 | 0.00 | -5.30 | 73.5 | 0.03 | 40.5 | 30.7 | 0.27 | 39.6 | 17.5 | 0.49 |
| C9 | -3.88 | 21.2 | 0.06 | -0.63 | 21.5 | 0.00 | 18.6 | 71.2 | 0.09 | 57.0 | -108 | 0.61 |
| C10 | -2.00 | 14.5 | 0.02 | -1.55 | 18.3 | 0.02 | 4.39 | 95.7 | 0.05 | 36.2 | -48.8 | 0.53 |
| C11 | 0.11 | 2.28 | 0.00 | -0.18 | 3.25 | 0.01 | -0.33 | 27.4 | 0.02 | 12.6 | -25.2 | 0.57 |
| C12 | -1.25 | 5.23 | 0.11 | -0.17 | 5.84 | 0.00 | 2.17 | 30.1 | 0.02 | 20.7 | -47.3 | 0.56 |
| C13 | -0.10 | 1.87 | 0.00 | 0.01 | 2.84 | 0.00 | -4.36 | 46.7 | 0.09 | 12.6 | -12.7 | 0.58 |
| C14 | -0.23 | 2.48 | 0.02 | 0.11 | 3.60 | 0.00 | 6.37 | 19.5 | 0.18 | 9.57 | -0.65 | 0.49 |
| C15 | -0.33 | 2.57 | 0.01 | -0.69 | 5.25 | 0.02 | 6.59 | 57.0 | 0.12 | 18.3 | 5.08 | 0.43 |
| C16 | 0.85 | 0.81 | 0.09 | -0.01 | 2.75 | 0.00 | 5.57 | 29.6 | 0.08 | 8.18 | 9.85 | 0.28 |
| C17 | -0.35 | 1.66 | 0.04 | 0.05 | 1.33 | 0.00 | 4.73 | 4.67 | 0.31 | 5.79 | 0.94 | 0.35 |
| C18 | -6.42 | 25.2 | 0.10 | -1.90 | 21.7 | 0.01 | 0.81 | 6.19 | 0.03 | 8.61 | -26.2 | 0.44 |
| C19 | -2.80 | 7.88 | 0.15 | -0.35 | 3.38 | 0.01 | 3.74 | 4.81 | 0.50 | 7.08 | -9.39 | 0.38 |
| C20 | 0.11 | 1.33 | 0.00 | -1.04 | 5.44 | 0.07 | 0.79 | 2.18 | 0.14 | 2.38 | -3.11 | 0.34 |
| Average | | | 0.05 | | | 0.01 | | | 0.15 | | | 0.47 |
| Min | -6.42 | 0.81 | 0.00 | -5.30 | 1.33 | 0.00 | -4.36 | 2.18 | 0.02 | 2.38 | -108 | 0.28 |
| Max | 0.85 | 34.3 | 0.15 | 0.11 | 73.5 | 0.07 | 40.5 | 95.7 | 0.50 | 57.02 | 17.5 | 0.61 |

Formatted Table

$C_{\text{n-alkanals}} = m\, C_{NOx} + b$

| Carbon atoms | EL | | | MR | | | | | |
|---|---|---|---|---|---|---|---|---|---|
| | | | | Northerly Winds | | | Southerly winds | | |
| | m | b | r$^2$ | m | b | r$^2$ | m | b | r$^2$ |
| C8 | -0.75 | 72.9 | 0.12 | 0.67 | 27.4 | 0.21 | 0.53 | 32.8 | 0.50 |
| C9 | 1.05 | 4.38 | 0.31 | 0.38 | 54.4 | 0.06 | 0.56 | -5.49 | 0.48 |
| C10 | 0.44 | 10.4 | 0.16 | 0.11 | 83.2 | 0.01 | 0.36 | 13.0 | 0.42 |
| C11 | 0.16 | 1.48 | 0.13 | 0.01 | 25.6 | 0.01 | 0.11 | 1.73 | 0.49 |
| C12 | 0.06 | 5.51 | 0.01 | 0.14 | 13.1 | 0.19 | 0.18 | -2.49 | 0.45 |
| C13 | 0.08 | 2.92 | 0.07 | -0.02 | 35.3 | 0.01 | 0.14 | 1.35 | 0.38 |
| C14 | 0.03 | 3.59 | 0.02 | 0.15 | 12.1 | 0.23 | 0.13 | 4.64 | 0.35 |
| C15 | 0.00 | 6.14 | 0.00 | 0.20 | 39.9 | 0.15 | 0.14 | 47.4 | 0.15 |
| C16 | 0.07 | 1.72 | 0.07 | 0.18 | 17.0 | 0.20 | 0.11 | 11.0 | 0.12 |
| C17 | -0.09 | 7.04 | 0.05 | 0.08 | 4.19 | 0.28 | 0.05 | 11.4 | 0.11 |
| C18 | -0.49 | 34.1 | 0.09 | 0.02 | 5.64 | 0.03 | 0.07 | -8.65 | 0.30 |
| C19 | 0.05 | 6.24 | 0.02 | 3.74 | 4.81 | 0.50 | 0.07 | 1.14 | 0.23 |
| C20 | -0.06 | 6.33 | 0.03 | 0.00 | 3.28 | 0.01 | 0.03 | -2.08 | 0.24 |
| Average | | | 0.08 | | | 0.15 | | | 0.32 |
| Min | -0.75 | 1.48 | 0.00 | -0.02 | 3.28 | 0.01 | 0.03 | -8.65 | 0.11 |
| Max | 1.05 | 72.9 | 0.31 | 3.74 | 83.2 | 0.50 | 0.56 | 47.4 | 0.50 |

Table S5.  Concentrations of $PM_{10}$ at the sampling sites.

| Sites | $PM_{10}$ Range, µg/m³ | $PM_{10}$ Mean µg/m³ | Note |
|---|---|---|---|
| RU and WM | 10.8-72.4 | 34.1 | The sampling period was dominated by southerly winds and the data from London, North Kensington were used as this is an upwind urban background site. |
| EL | 4.37-27.1 | 19.3 | The $PM_{10}$ data was obtained from the London North Kensington site (Defra), because the EL only have $PM_{2.5}$ data, and the $PM_{2.5}$ data of two site (EL and London North Kensington) were close to each other. |
| MR | 12.6-78.7 | 30.7 | MR site |

Table S4S6. Analysis of n-alkanals, alkan-2-ones and alkan-3-ones partitioning, all compounds, daily data at RU, WM, El and MR sites.

[revised manuscript text omitted]

| Sites | PM$_{10}$ Range, μg/m³ | PM$_{10}$ Mean μg/m³ | Note |
|---|---|---|---|
| RU and WM | 10.8-72.4 | 34.1 | The sampling period was dominated by southerly winds and the data from London, North Kensington were used as this is an upwind urban background site. |
| EL | 4.37-27.1 | 19.3 | The PM$_{10}$ data was obtained from the London North Kensington site (Defra), because the EL only have PM$_{2.5}$ data, and the PM$_{2.5}$ data of two site (EL and London North Kensington) were close to each other. |
| MR | 12.6-78.7 | 30.7 | MR site |